# Amortized Bayesian Causal Discovery of Extended Factor Graphs

## Abstract

Learning causal graphs from interventional data is a challenging problem with broad applications. In molecular biology, for example, a central goal is to uncover gene regulatory networks from large-scale perturbation data. An ideal algorithm for this task should scale to thousands of nodes, incorporate interventions even when their targets are unknown, quantify uncertainty, and provide identifiability guarantees. However, existing approaches—e.g. approaches using score-based optimization or approximate Bayesian inference—often fail to meet all of these criteria. To address these limitations, we develop Amortized Bayesian Causal Discovery of Extended Factor Graphs (ABCDEFG). Our method guarantees exact acyclicity, scales to graphs with thousands of nodes, and naturally handles interventions even when their targets are unknown. Additionally, ABCDEFG estimates a posterior distribution whose mode provably identifies the true causal graph up to an equivalence class. On simulated datasets, ABCDEFG achieves state-of-the-art accuracy, producing a well-calibrated posterior distribution while outperforming previous score-based and approximate Bayesian methods. Applied to large-scale single-cell perturbation data, ABCDEFG identifies both established and novel gene targets of growth factors [2].

## 1 Introduction

Discovering causal relationships is a fundamental challenge across scientific domains. In many settings, both observational and interventional data are available to probe underlying causal mechanisms. Yet, inferring causal relationships remains difficult in large, complex systems. For example, in computational biology, understanding how genes influence one another through gene regulatory networks is crucial for understanding cellular development and homeostasis. Recent biotechnological advances now enable high-throughput perturbation experiments, providing measurements of gene expression across thousands to millions of cells under various interventions, providing exciting new data for inferring causal relationships in the cell.

However, existing causal discovery methods fall short when applied to inferring a gene regulatory network from high-throughput perturbation data. Many approaches cannot scale to the large number of variables in the gene regulatory network (more than 20,000 genes) or the large number of samples ($10^4 - 10^6$ cells). Very noisy data, correlated causal edge probabilities, and interventions with unknown targets (such as drug treatments) pose additional challenges. While approximate Bayesian methods offer the advantage of uncertainty quantification (a crucial property for noisy biological data), they typically struggle to scale to problems of this size. Although prior work has addressed some of these issues in isolation, no existing method satisfies all the requirements simultaneously. There remains a need for new causal inference approaches that are scalable, uncertainty-aware, and capable of jointly learning causal gene relationships and intervention targets from large-scale single-cell drug or growth factor screens.

To address these challenges, we develop **A**mortized **B**ayesian **C**ausal **D**iscovery of **E**xtended **F**actor **G**raphs (ABCDEFG). Our key idea is to represent causal structures using *extended factor graphs*, where feature nodes and intervention nodes are connected through auxiliary factor nodes. This extended factor graph formulation enables accurate and scalable distributional estimation of causal DAGs, while incorporating interventions with unknown targets and guaranteeing acyclicity. Moreover, it supports joint modeling of edge probabilities as coupled random variables, capturing complex

Table 1: Summary of the proposed and existing approaches. Max nodes and samples indicate the size of the largest dataset evaluated in the original publication.

| Method | DAG Uncertainty | Graph Model Size | Guaranteed Acyclic | Intvn Data | Unknown Target | Max Nodes | Max Samples |
|---|---|---|---|---|---|---|---|
| NO-TEARS | ✗ | $O(n^2)$ | ✗ | ✗ | ✗ | 100 | 7,466 |
| DCDI | ✗ | $O(n^2)$ | ✗ | ✓ | ✓ | 100 | $10^6$ |
| DAGMA | ✗ | $O(n^2)$ | ✗ | ✗ | ✗ | 2,000 | 1,000 |
| DCD-FG | ✗ | $O(mn)$ | ✗ | ✓ | ✗ | 1,000 | 87,590 |
| ENCO | ✗ | $O(n^2)$ | ✗ | ✓ | ✗ | 1,000 | 110,000 |
| SDCD | ✗ | $O(n^2)$ | ✗ | ✓ | ✗ | 4,000 | 10,500 |
| DeepITE | ✗ | $O(n^2)$ | ✗ | ✓ | ✓ | 500 | 10,000 |
| LIT | ✗ | $O(n^2)$ | ✗ | ✓ | ✓ | 16 | 32 |
| iSCAN | ✗ | $O(n^2)$ | ✗ | ✓ | ✓ | 50 | 1,000 |
| BaCaDI | ✓ | $O(n^2)$ | ✗ | ✓ | ✓ | 20 | 300 |
| ProDAG | ✓ | $O(n^2)$ | ✓ | ✗ | ✗ | 100 | 7,466 |
| DECI | ✓ | $O(n^2)$ | ✗ | ✗ | ✗ | 64 | 5,000 |
| DP-DAG | ✓ | $O(n^2)$ | ✓ | ✗ | ✗ | 100 | 1,000 |
| VDESP | ✓ | $O(n^2)$ | ✓ | ✗ | ✗ | 20 | 4,200 |
| ABCDEFG (ours) | ✓ | $O(mn)$ | ✓ | ✓ | ✓ | 1,000 | 31,425 |

dependencies among edges. ABCDEFG also possesses strong theoretical guarantees: we prove that the mode of the estimated posterior recovers the true causal graph up to an equivalence class.

**Contributions.** Our core contributions include: (1) we introduce a new parametric model for sampling extended factor graphs that are acyclic by construction and have explicit intervention nodes; (2) we develop a variational Bayesian approach for discovering causal extended factor graphs from interventional data with known or unknown targets; (3) we integrate sum-product networks into the generative model to flexibly model complex joint distributions over causal edges; (4) we develop new theoretical results connecting our Bayesian framework to the identifiability guarantees of score-based methods; and (5) we demonstrate the effectiveness of ABCDEFG on a large-scale single-cell perturbation dataset, recovering both known and novel gene-to-gene and growth factor-to-gene interactions.

**Related Work.** Classical causal discovery methods are typically divided into constraint-based and score-based methods. Constraint-based methods date back to the 90s when Spirtes & Glymour [24] proposed the PC algorithm. In contrast, score-based differentiable causal discovery methods have gained popularity in recent years due to their better performance and computational efficiency. Zheng et al. [31] pioneered the formulation of causal DAG discovery as a continuous optimization problem under a linear causal model, using an augmented Lagrangian approach with a matrix exponential constraint to enforce acyclicity. Lee et al. [15] built on this by designing a polynomical regression loss tailored and reducing computational cost for gene expression data. Subsequent works improved performance and expanded the modeling framework. Bello et al. [4] proposed an alternative log-det function for the acyclicity constraint, resulting in better performance, better-behaved gradient and faster convergence. Lippe et al. [16] designed an optimization strategy alternating between distribution and graph fitting and proved convergence to the true graph under specific conditions.

A parallel line of work developed Bayesian methods for causal discovery. Cundy et al. [9] applied variational inference (VI) to linear Gaussian SEMs. Annadani et al. [3] adopted the NoCurl DAG model [30] and derived a VI method for the parameters. Charpentier et al. [7] proposed a fully probabilistic and differentiable DAG model and performs VI by maximizing the ELBO. Geffner et al. [10] developed a Bayesian method based on a previous probabilistic DAG model [16] and applied a flow-based generative model for distributional fitting. Thompson et al. [26] proposed a Bayesian method for DAGs by first pruning a weighted matrix to be acyclic and projecting it onto an L1 ball. Bonilla et al. [5] designed a differentiable DAG distribution using a continuous relaxation of permutation [21]. These Bayesian methods tend to be significantly less scalable than the score-based methods, as reflected in the relatively small datasets used for evaluation.

The methods discussed above focus exclusively on observational data and are not designed to incorporate interventional data, which is critical for accurate causal discovery in applications such as computational biology. To address this, a separate line of work has explored causal discovery with interventions. Brouillard et al. [6] proposed a differentiable method that incorporates observational and interventional data; guarantees identifiability with known or unknown intervention targets; and model nonlinear effects using deep neural networks. Lopez et al. [17] used factor graphs to learn a low-rank approximation of DAGs, a key foundation for our approach. Nazaret et al. [18] proposed a robust acyclicity penalty loss. Hägele et al. [11] set up a Bayesian framework for causal discovery with interventional data. Our work is also distinct from intervention target estimation methods, which can infer the nodes targeted by interventions but cannot simultaneously estimate the causal graph (e.g., iSCAN [8], LIT [28], and DeepITE [25]). We summarize these and related methods, along with our own, in Table 1.

## 2 METHODS

### 2.1 DEFINITIONS

Our definitions and notation closely parallel previous differentiable causal discovery methods [6], but we summarize the key points here to make the presentation of our approach more self-contained. Let $X = \{X_1, \ldots, X_n\}$ be a set of random variables. A causal graphical model (CGM) for these variables consists of a joint distribution and a graph $\{G = (V, E), p(X)\}$. $G \in \mathcal{G}$ (where $\mathcal{G}$ is the set of DAGs) and $G$ and $p$ are related as follows:

$$p(X) = \prod_{i \in V} p(X_i | X_{\pi_i})$$

Here, $\pi_i$ is the set of parents of vertex $i$ in $G$. Intuitively, an intervention on a variable modifies its conditional dependence on its parent. Interventions can be performed on multiple variables simultaneously; the *interventional target* for each intervention is thus a set of vertices $I \subset V$.

Given a CGM with $\{G, p(X)\}$, intervening on targets $I$ modifies $p$ into $p^I$:

$$p^I(X) = \prod_{i \in I} p^I(X_i | X_{\pi_i}) \prod_{i \notin I} p(X_i | X_{\pi_i})$$

Note that the causal sufficiency assumption is implicit in this definition of intervention. The $I$-faithfulness assumption ensures that $p^I(X_i | X_{\pi_i}) \neq p(X_i | X_{\pi_i})$. A *hard intervention* removes all dependence on parents, so $p^{I_k}(X_i | X_{\pi_i}) = p^{I_k}(X_i)$.

To accommodate multiple interventions, we define an *intervention set* as $\mathcal{I} := (I_1, \ldots, I_{n^{\mathcal{I}}})$, where $n^{\mathcal{I}}$ is the number of interventions. Note that the intervention set may include multiple interventions with the same targets, $I_j = I_k$. For convenience, we include the observational distribution in the intervention set and define it as $I_1 := \emptyset$. We also abbreviate $p^{I_k}(X)$ as $p^{(k)}(X)$. The set of joint distributions induced by a causal graph and intervention set is $\mathcal{M}_{\mathcal{I}^*}(G)$, which we can factorize according to the Markov property: $\mathcal{M}_{\mathcal{I}^*}(G) := \{p^{I_k}(X) = \prod_{i=1}^n p^{I_k}(X_i | X_{\pi_i})\}$

Our goal is to estimate $q(G; \Lambda)$, a probability mass function (PMF) over $\mathcal{G}$ parameterized by a set of real numbers $\Lambda$. In estimating $q(G; \Lambda)$, we will make use of $f(X; \Phi)$ and $f^I(X; \Phi)$, density models of $p(X)$ and $p^{(k)}(X)$, respectively, parameterized by a set of real numbers $\Phi$.

### 2.2 FACTOR DIRECTED ACYCLIC GRAPHS (F-DAGS)

Our goal is to build a generative model for DAGs and ultimately a Bayesian framework for inferring causal DAGs. To do this, we start with a type of graph called a factor DAG (f-DAG), following Lopez et al. [17]. An f-DAG is formally defined as follows:

**Definition 2.1** (Lopez et al. [17])**.** Given a set of nodes, $V$, and factors, $F$, a factor directed acyclic graph (f-DAG), denoted as $(V, F, E)$, is a directed acyclic graph $(V \cup F, E)$ where edges $E \subset \{(i, j) : i \in V, j \in F \text{ or } i \in F, j \in V\}$.

Given an f-DAG, we can preserve the connection between any two nodes (factors) by removing all intermediate factors (nodes) along paths. This results in a node-only (factor-only) graph:

**Definition 2.2** (Lopez et al. [17]). Given an f-DAG, $D = (V, F, E)$, its half-square node graph is defined as $D^2[V] = (V, \{(i,j) : \exists f \in F, (i,f), (f,g) \in E\})$, and half-square factor graph is defined as $D^2[F] = (F, \{(f,g) : \exists i \in V, (f,i), (i,g) \in D\})$.

Let $\mathbf{A}$ be the adjacency matrix of a causal DAG. An f-DAG can be viewed as a Boolean factorization of $\mathbf{A}$, $\mathbf{A} = \mathbf{UV}$. Here $\mathbf{U} \in \{0,1\}^{n \times m}$ and $\mathbf{V} \in \{0,1\}^{m \times n}$ are binary node-to-factor and factor-to-node connection matrices. Intuitively, if $m < n$, the node-only half-square graph of an f-DAG can be interpreted as a low-rank approximation of the full-rank DAG, and the factors represent groups of related nodes (modules, topics, etc.). Lopez et al. [17] proved that, with probability exponentially approaching one, adding incorrect edges to a random graph increases its Boolean rank. Viewing an f-DAG as a Boolean matrix factorization of the binary adjacency matrix (Fig. 1), this result implies that the low-rank property of the f-DAG acts as a regularization for graph structure and increases robustness to noisy edges. This low-rank assumption is common in computational biology [29; 32].

We further extend the f-DAG framework for identifying unknown intervention targets. We model the effect of each intervention on target nodes via factors. This is a natural abstraction for interventions whose exact targets are unknown, such as drugs that affect a biological pathway. Suppose $\mathcal{I} = \{I_1, \ldots, I_{n^{\mathcal{I}}}\}$ is a set of unknown intervention targets, and $\mathbf{W}$ is a $n^{\mathcal{I}}$-by-$m$ binary matrix, where $W_{kj}$ represents whether the $k$-th intervention targets the $j$-th factor. We next define extended f-DAGs, a.k.a. extended factor graphs.

**Definition 2.3** (Extended f-DAG). Let $D = (V, F, E)$ be an f-DAG and $\mathcal{I} = \{I_1, \ldots, I_{n^{\mathcal{I}}}\}$ be a set of interventions. Let $\Xi = \{\xi_k, k \in [n^{\mathcal{I}}]\}$ be $n^{\mathcal{I}}$ nodes corresponding to the $n^{\mathcal{I}}$ interventions. An extended f-DAG is defined as an f-DAG $D^{\mathcal{I}} = (V \cup \Xi, F, E \cup E^{\mathcal{I}})$ where $E^{\mathcal{I}} \subseteq \{(\xi_k, l) : l \in F\}$, i.e. set of edges from intervention nodes to factors.

### 2.3 Probabilistic Modeling of f-DAGs

**Generative Model for f-DAGs.** A key innovation of our approach is a generative process for efficiently sampling large-scale f-DAGs that guarantees acyclicity by construction. This eliminates the need for computationally expensive acyclicity penalties used in differentiable causal discovery methods, ensures that all sampled graphs are acyclic, and forms the foundation for probabilistic causal f-DAG inference.

Given a set of $n$ nodes, $\{v_i : i \in [n]\}$, and $m$ factors, $\{f_j : j \in [m]\}$, we construct an f-DAG by forming a partial order of nodes and factors together and determining the node-to-factor or factor-to-node edge connection (Fig. 1). Since node-to-node edges are disallowed in f-DAGs (nodes are only connected via factors), we do not need to explicitly model the relative order between nodes. Instead, we form a total order of factors, $\tau : [m] \to [m]$, such that $f_{\tau(1)} < \ldots < f_{\tau(m)}$. They partition all nodes into $m + 1$ subsets and each node $v_i$ is randomly inserted into one partition, i.e. $\exists k \in [m], f_{\tau(k-1)} < v_i < f_{\tau(k)}$ or $v_i < f_{\tau(1)}$ or $v_i > f_{\tau(m)}$. We model this assignment using $n$ categorical distributions with $m + 1$ categories, denoted as $\mathbf{Y} = \{Y_i : i \in [n]\}$. The second step determines edge *existence*, regardless of direction. These edge connection probabilities are related to a joint distribution of all edge connections. We use a binary matrix $\mathbf{B} \in \{0,1\}^{n \times m}$ to represent edge connections. Thus, $\mathbf{Y}$ contains all the direction information and $\mathbf{B}$ contains all the connection information. Hence, $\mathbf{Y}$ and $\mathbf{B}$ uniquely determine an f-DAG, and we can generate an f-DAG by sampling $\mathbf{Y}$ and $\mathbf{B}$ (Fig. 1).

**Sampling Independently or Jointly Distributed Causal Edges.** Using the above generative process, we can infer a causal DAG by optimizing a score function with respect to $\mathbf{Y}$ and $\mathbf{B}$. But what is the best way to sample $\mathbf{Y}$ and $\mathbf{B}$? One possibility is to model the edges as independent Bernoulli random variables sampled using the Gumbel softmax trick [13]. However, such a naive approach neglects possible correlation between edges. A more general approach is to model the joint distribution of edges using a sum-product network (SPN) [20; 23]. SPNs combine sum and product operations over latent variables, enabling flexible sampling from a categorical joint distribution (see Appendix A for further details). We implemented and evaluated both strategies on real and simulated data.

### 2.4 Bayesian Causal Discovery of DAGs

**A Differentiable Bayesian Framework for Causal Discovery.** Let $\mathcal{G}$ be the set of all DAGs. Consider a generative process where a DAG is first sampled from a prior, $p(G)$ with support on $\mathcal{G}$,

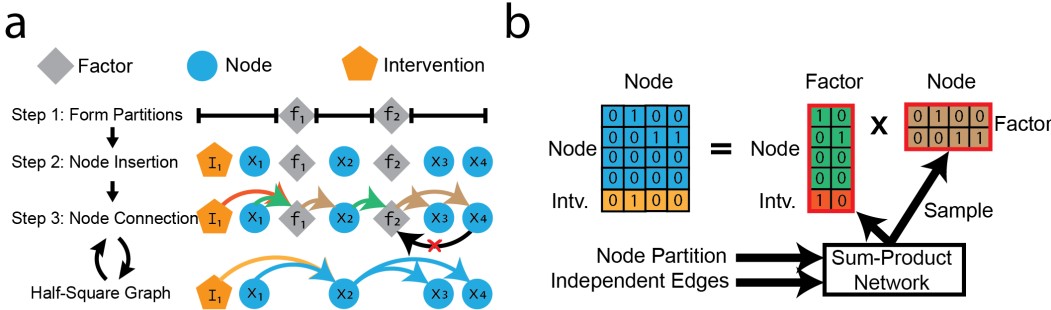

Figure 1: **Causal inference using extended factor graphs. (a)** Generative process for sampling extended factor graphs that are guaranteed to be acyclic. Factors are ordered to form partitions, then nodes and interventions are inserted into partitions. Finally, edges are added from earlier nodes, factors or interventions to later. Removing factors gives a "half-square" graph with direct node-to-node and intervention-to-node connnections. **(b)** An extended factor graph factorizes a node/intervention-to-node adjacency matrix as a Boolean product of a node/intervention-to-factor and factor-to-node matrix. ABCDEFG samples edges in these matrices using either independent Bernoulli random variables or a joint PMF parametrized by a sum-product network.

and a generative model $p(\boldsymbol{X}|G, I)$ under the intervention $I$. Given empirical observations, we can obtain a MAP estimate of the causal graph as $G^* = \arg\max_{G \in \mathcal{G}} p(G|\boldsymbol{X}, I)$.

Because $|\mathcal{G}|$ is super-exponential in $n$ [22], searching through the discrete space is computationally inefficient for large $n$. Instead, we resort to continuous optimization. As the true posterior is often intractable, we apply variational Bayes using a variational distribution $q(G; \boldsymbol{\Lambda})$. In this way, we are able to find $G^*$ by optimizing a KL divergence: $G^* = \arg\min_{G \in \mathcal{G}} KL(q(G; \boldsymbol{\Lambda})||p(G|\boldsymbol{X}, I))$. In real experimental scenarios, the random intervention is replaced with Monte Carlo sampling, $I_1, \ldots, I_{n^{\mathcal{I}}}$. From our derivation (Appendix B.2), minimizing the KL divergence is equivalent to maximizing the evidence lower bound (ELBO):

$$q^*(G) = \underset{q(G;\boldsymbol{\Lambda})}{\arg\max} \sum_{k=1}^{n^{\mathcal{I}}} \mathbb{E}_{p^{(k)}(\boldsymbol{X}|G^*)} \left[ \mathbb{E}_{q(G;\boldsymbol{\Lambda})} \left[ \log p_\Phi^{(k)}(\boldsymbol{X}|G) \right] \right] - KL\left( q(G;\boldsymbol{\Lambda})||p(G) \right). \quad (1)$$

This ELBO objective is directly connected to autoencoding variational Bayes [14]. A slight difference compared to the traditional autoencoding variational Bayes setting is that we treat the causal graph as a constant during the likelihood calculation, so the expectation is over $p^{(k)}(\boldsymbol{X}|G^*)$ instead of $p^{(k)}(\boldsymbol{X})$. (We provide a detailed derivation of the ELBO in the Appendix.) The posterior can be estimated by optimizing the ELBO to yield $q^*(G) = p_\Phi^{(k)}(G|\boldsymbol{X})$, assuming enough capacity of the variational family.

As mentioned in Section 2.2, we can narrow down the search space by considering extended f-DAGs as a reasonable low-rank approximation of the true causal DAG. In this work, we use either independent Bernoullis or SPNs as a parametric model for f-DAGs, but the Bayesian framework is general to parametric DAG models.

## 2.5 AMORTIZED BAYESIAN CAUSAL DISCOVERY OF EXTENDED FACTOR GRAPHS

With the problem setup in Section 2.4, we now formally introduce our method, Amortized Bayesian Causal Discovery of Extended Factor Graphs (ABCDEFG). (Note that "amortized" here refers to using a common inference function in contrast to traditional mean-field variational inference. Variational autoencoders (VAEs) are a type of amortized variational inference [1].) Given a set of random variables $\boldsymbol{X} = \{X_i : i \in [n]\}$ generated via a causal graph $G^*$, we apply a Bayesian method by estimating $p(G|\boldsymbol{X}, I^*)$ via optimization as described in section 2.4:

$$q^*(G) = \underset{q(G;\boldsymbol{\Lambda})}{\arg\max} \sum_{k=1}^{n^{\mathcal{I}}} \mathbb{E}_{p^{(k)}(\boldsymbol{X}|G^*)} \left[ \mathbb{E}_{q(G;\boldsymbol{\Lambda})} \left[ \log p_\Phi^{(k)}(\boldsymbol{X}|G) \right] \right] - KL\left( q(G;\boldsymbol{\Lambda})||p(G) \right).$$

The key to convert discrete search into continuous optimization is thus to create a differentiable parametric model for DAGs and estimate the ELBO using Monte Carlo sampling. We assume the true causal graph is or can be approximated by an f-DAG. Thus, we use either independent Bernoullis sampled by Gumbel softmax or joint PMF sampled from an SPN to parameterize $q(G; \mathbf{\Lambda})$.

The model architecture (bottom panel of Fig. 4) consists of an f-DAG parametric model (Gumbel softmax or SPN) and a VAE for data distribution fitting. The output is a node-to-factor matrix $\mathbf{U} \in \mathbb{R}^{n \times m}$ and a factor-to-node matrix $\mathbf{V} \in \mathbb{R}^{m \times n}$. Next, we model the data distribution under the f-DAG as $p(\mathbf{X}) = \int_{\mathbf{Z}} \prod_{i=1}^{n} f(Z_j | \mathbf{X}_{\pi_j}) g(X_i | \mathbf{Z}_{\pi_i}) d\mathbf{Z}$. Here, $\pi_i$ and $\pi_j$ are the parent nodes and factors in the f-DAG. Instead of using separate encoding and decoding functions to obtain the posterior of each $Z_j$ and conditional likelihood of each $X_i$, we follow Lopez et al. [17] and amortize all conditional distributions into a single encoding and decoding feed-forward neural network. Causal relations are injected into the VAE via masking operations $\mathbf{U_j} \odot \mathbf{X}$ and $\mathbf{V_i} \odot \mathbf{Z}$, where $\mathbf{U_j}$ is the $j$-th column of $\mathbf{U}$, $\mathbf{V_i}$ is the $i$-th column of $\mathbf{V}$ and $\odot$ denotes the Hadamard product.

When the intervention targets are unknown, the causal discovery problem can be treated as recovering an extended f-DAG with intervention nodes. Equivalently, our Gumbel softmax or SPN sampling procedure can be extended to generate an intervention-to-factor matrix $\mathbf{W} \in \{0, 1\}^{k \times m}$. The causal mask operation becomes $[\mathbf{U_j} \odot \mathbf{X}; \mathbf{W_j} \odot \mathbf{I}]$ where $\mathbf{I}$ is a one-hot encoding of the intervention. We can apply the same optimization approach to jointly infer the causal graph and intervention targets. Extended f-DAGs could also include intervention information such as the dosage of a chemical treatment, though we did not explore this in detail here.

## 2.6 IDENTIFIABILITY

We next provide identifiability guarantees for our approach. Our main theorem proves that the DAG with highest posterior probability (MAP estimate) belongs to the same equivalence class as the true causal DAG. We use the notion of $\mathcal{I}$-Markov equivalence from [6]: two DAGs $G_1$ and $G_2$ are $\mathcal{I}$-Markov equivalent if and only if $\mathcal{M}_{\mathcal{I}}(G_1) = \mathcal{M}_{\mathcal{I}}(G_2)$. Our theorem relies on the same four assumptions as previous identifiability results for differentiable causal inference methods [6]: sufficient model capacity, $\mathcal{I}$-faithfulness, positivity, and finite differential entropy. This result applies to any DAG, including half-square graphs obtained from f-DAGs.

**Theorem 2.4** (Identifiability via ELBO maximization). *Let $\mathbf{X}$ be a set of causally related random variables with a causal DAG $G^*$ and $\mathcal{I}^*$ be a set of interventions with $I_1^* = \emptyset$. Let $\mathcal{G}$ be a subset of all causal DAGs and $q^*(G)$ be an optimal graph distribution from the optimization problem:*

$$\sup_{q(G; \mathbf{\Lambda}): supp(q) \subseteq \mathcal{G}} \mathcal{L}(q(G; \mathbf{\Lambda})),$$

*where*

$$\mathcal{L}(q(G; \mathbf{\Lambda})) = \mathbb{E}_{q(G; \mathbf{\Lambda})}[S_{\mathcal{I}^*}(G)] - \beta KL(q(G; \mathbf{\Lambda}) || p(G)), \; \beta > 0,$$

$$S_{\mathcal{I}^*}(G) = \sup_{\mathbf{\Phi}} \sum_{k=1}^{n^{\mathcal{I}^*}} \mathbb{E}_{p^{(k)}(\mathbf{X})} \left[ \log f^{(k)}(\mathbf{X} | G; \mathbf{\Phi}) \right] - \lambda |G|.$$

*In addition, assume the following:*

1. *Sufficient capacity: The set of distributions from our parametric models contains the ground truth interventional distributions: $\{p^{(k)}(\mathbf{X}) : k \in [n^{\mathcal{I}^*}]\} \in \mathcal{F}_{\mathcal{I}^*}(G^*)$ where $\mathcal{F}_{\mathcal{I}^*}(G^*) = \{\{f^{(k)}(\mathbf{X} | G^*; \mathbf{\Phi})\} : \mathbf{\Phi} \in \Omega(\mathbf{\Phi})\}$.*

2. *$\mathcal{I}$-faithfulness as defined in [6] (See appendix B, Thm. B.13 for details).*

3. *Positivity: $\forall G, I, \mathbf{\Phi}, f^{(k)}(\mathbf{X} | G, I; \mathbf{\Phi}) > 0$.*

4. *Finite differential entropy: $\forall k \in [n^{\mathcal{I}^*}], \left| \mathbb{E}_{p^{(k)}(\mathbf{X})} \left[ \log p^{(k)}(\mathbf{X}) \right] \right| < +\infty$.*

*If $G^* \in \mathcal{G}$, then, under the assumptions 1-4 [6] and with a proper $\beta > 0$, $\hat{G} = \arg \max_G q^*(G)$ is $\mathcal{I}^*$-Markov equivalent to $G^*$.*

Table 2: F1 score and SHD of Scored methods on Nonlinear Targeted Simulated Datasets

| METRIC | METHOD | HARD INTVN | SOFT INTVN | SPN HARD | SPN SOFT |
|---|---|---|---|---|---|
| F1 | DCDI | $0.19 \pm 0.05$ | $\underline{0.25 \pm 0.07}$ | $0.34 \pm 0.01$ | $0.35 \pm 0.04$ |
| | DCDFG | $0.05 \pm 0.08$ | $0.20 \pm 0.14$ | $0.23 \pm 0.18$ | $0.57 \pm 0.14$ |
| | ENCO | $0.10 \pm 0.01$ | $0.10 \pm 0.03$ | $0.25 \pm 0.01$ | $0.23 \pm 0.03$ |
| | SDCD | $\mathbf{0.31 \pm 0.01}$ | $\mathbf{0.30 \pm 0.06}$ | $0.25 \pm 0.02$ | $0.30 \pm 0.06$ |
| | ABCDEFG | $\underline{0.29 \pm 0.03}$ | $\underline{0.25 \pm 0.01}$ | $\mathbf{0.64 \pm 0.01}$ | $\mathbf{0.61 \pm 0.03}$ |
| | ABCDEFG (SPN) | $\underline{0.29 \pm 0.04}$ | $\underline{0.21 \pm 0.01}$ | $\underline{0.61 \pm 0.02}$ | $\underline{0.60 \pm 0.02}$ |
| SHD | DCDI | $\underline{740 \pm 291}$ | $\underline{559 \pm 106}$ | $4293 \pm 301$ | $3337 \pm 120$ |
| | DCDFG | $2513 \pm 0$ | $900 \pm 272$ | $2500 \pm 198$ | $\mathbf{2030 \pm 125}$ |
| | ENCO | $1952 \pm 126$ | $1992 \pm 141$ | $2855 \pm 177$ | $2896 \pm 100$ |
| | SDCD | $\mathbf{421 \pm 77}$ | $\mathbf{421 \pm 78}$ | $2973 \pm 72$ | $2793 \pm 83$ |
| | ABCDEFG | $1114 \pm 328$ | $1406 \pm 361$ | $\mathbf{2046 \pm 49}$ | $2248 \pm 200$ |
| | ABCDEFG (SPN) | $1125 \pm 248$ | $1791 \pm 249$ | $\underline{2206 \pm 81}$ | $\underline{2228 \pm 85}$ |

The key idea of the proof is that any posterior distribution whose MAP is not $\mathcal{I}^*$-Markov equivalent to the true causal DAG must have a lower ELBO. Here, we present a sketch proof. See Appendix B.2 for details.

*Proof.* The proof is by contradiction. Suppose $\exists \hat{G} = \arg\max_G q^*(G)$ that is not $\mathcal{I}^*$-Markov equivalent to $G^*$. We can create another distribution $q'$ such that $q'(\hat{G}) - q^*(\hat{G}) = q^*(G^*) - q'(G^*) = \epsilon > 0$ and for any other graph $G$, $q'(G) = q^*(G)$. From algebraic calculation, we have

$$\mathcal{L}(q') - \mathcal{L}(q^*) = \epsilon \left( S_{\mathcal{I}^*}(G^*) - S_{\mathcal{I}^*}(\hat{G}) \right) + \beta \Delta.$$

Because $S_{\mathcal{I}^*}(G^*) - S_{\mathcal{I}}(\hat{G}) > 0$, $\exists \beta > 0$ such that $\mathcal{L}(q') - \mathcal{L}(q^*) > 0$. Then, we have a contradiction about $q^*$ being an optimal solution to the optimization problem. $\square$

Furthermore, our method can be extended to identify the true causal DAG by replacing the causal DAG with an interventional DAG ($\mathcal{I}$-DAG)[27]. Because the derivation is highly similar to that of causal discovery with known targets, we present the derivation of the ELBO objective and identifiability results in Appendix Section B.3.

## 3 EXPERIMENTS

### 3.1 SIMULATION RESULTS

We simulated data based on the approach of [17]. We further explored the effects of correlations between edge probabilities, which our approach explicitly models but previous approaches do not, by constructing an SPN and then sampling from the joint distribution of edges. We also simulated interventions with unknown targets. To evaluate our method, we benchmarked ABCDEFG on 24 datasets and compared with four SOTA score-based methods: DCDI [6], DCDFG [17], ENCO [16] and SDCD [18]. The 24 datasets include eight types of SEMs – a combination of (1) linear vs. non-linear causal effects, (2) independent vs. jointly distributed edge probabilities, and (3) hard vs. soft interventions. Each simulated graph includes 100 nodes and 10 factors. We simulated three separate graphs for each type of SEM. Similar to previous studies, we report Structural Hamming Distance (SHD) and F1 score for edge prediction. We used consistent hyperparameter settings for ABCDEFG across all simulations (Appendix C.3). ABCDEFG significantly outperformed all other approaches on graphs with nonlinear causal effects and edge probabilities that are jointly distributed and sampled from an SPN (Table 2). ABCDEFG performed similarly or better than SOTA methods on nonlinear SEMs, though SDCD showed strong performance in the nonlinear, non-SPN setting (Fig. 5). We also found that the other methods frequently produced cyclic graphs that required heuristic pruning to obtain a final DAG (Fig. 6, Fig. 7).

Table 3: F1 score and SHD of ABCDEFG on Nonlinear Untargeted Simulated Datasets

| METRIC | METHOD | HARD INTVN | SOFT INTVN | SPN HARD | SPN SOFT |
|---|---|---|---|---|---|
| F1 | ABCDEFG | $0.23 \pm 0.01$ | $0.23 \pm 0.05$ | $0.22 \pm 0.06$ | $0.46 \pm 0.03$ |
| | ABCDEFG (SPN) | $0.20 \pm 0.02$ | $0.17 \pm 0.02$ | $0.28 \pm 0.04$ | $0.55 \pm 0.05$ |
| | ABCDEFG INTV. | $0.36 \pm 0.01$ | $0.38 \pm 0.01$ | $0.46 \pm 0.10$ | $0.85 \pm 0.02$ |
| | ABCDEFG (SPN) INTV. | $0.35 \pm 0.01$ | $0.35 \pm 0.02$ | $0.51 \pm 0.01$ | $0.84 \pm 0.01$ |
| SHD | ABCDEFG | $857 \pm 112$ | $1121 \pm 261$ | $3067 \pm 56$ | $2632 \pm 217$ |
| | ABCDEFG (SPN) | $1076 \pm 326$ | $1399 \pm 342$ | $3132 \pm 148$ | $2307 \pm 239$ |
| | ABCDEFG INTV. | $1659 \pm 240$ | $1426 \pm 346$ | $2584 \pm 440$ | $1021 \pm 56$ |
| | ABCDEFG (SPN) INTV. | $1761 \pm 204$ | $1516 \pm 280$ | $2438 \pm 187$ | $1071 \pm 71$ |

We next evaluated how ABCDEFG performs for interventions with unknown targets, a key advantage of our approach. To test target identification, we generated causal graphs but withheld the intervention target information during inference. SDCD, ENCO, and DCDFG cannot incorporate interventions with unknown targets. Although DCDI can in principle identify both causal relations and unknown intervention targets, we excluded it from this evaluation because it required extremely long runtimes and showed poor performance in the easier targeted case. In addition to SHD and F1 of the causal graph, we evaluated the accuracy of the intervention-to-node graph (Table 3). The accuracy of inferred node-to-node relationships was lower compared to interventions with known targets, indicating that causal inference is more challenging under unknown interventions. Nevertheless, ABCDEFG inferred the intervention targets more accurately than the node-to-node causal relationships, achieving relatively high precision and recall, particularly for SPN-simulated graphs.

We also benchmarked ABCDEFG against SOTA Bayesian causal inference methods: BaCaDi [12], ProDAG [26], DECI [10] and VI-DP-DAG [7]. These methods required significantly longer runtimes than the score-based approaches, so we used smaller datasets with 16 nodes and 260 samples. ABCDEFG and ProDAG were significantly faster than the other Bayesian approaches (see Table 14). For each method, we sampled 100 graphs from the posterior after training. ABCDEFG outperformed the other methods by achieving the highest F1 score and the lowest SHD across four different linear and nonlinear settings (Table 4). We also evaluated the posterior calibration of each method by comparing the expected and predicted edge probabilities. The posterior estimated by ABCDEFG showed the best match between the predicted edge probability and empirical estimation (Fig. 2a).

## 3.2 APPLICATION TO REAL CELLULAR PERTURBATION SCREEN

We applied our model to a large-scale single-cell perturbation screen in which cells were treated with 46 combinations of 14 growth factors [2]. Growth factors are biomolecules that induce significant molecular changes through signaling pathways and are used to steer cells toward desired cell types in the dish. Though some downstream targets of growth factors are known, the targets are highly context-specific. The raw data contains gene expression counts for 34,469 genes in 31,475 cells. Following standard preprocessing steps for this type of data, we extracted the 1,000 most highly variable genes for causal graph inference. We used 10 factors in our model. To evaluate intervention target identification, we collected (growth factor,gene) pairs from the Gene Ontology and used these true positives to calculate recall. We cannot calculate precision because the full signaling network is unknown, so true negatives are not available. As a baseline model, we compared against random factor graphs with the same edge density as the graphs inferred by ABCDEFG. ABCDEFG achieved a recall of 0.325 (Basic) and 0.376 (SPN), significantly better than the baseline model (recall: 0.196). Second, we evaluated data reconstruction on held-out interventions. Both DCDI and ENCO failed to run on the real data. The remaining approaches DCDFG and SDCD cannot incorporate interventions with unknown targets, so we treated the data as observational when training them. We held out four

Table 4: F1 score and SHD of Bayesian methods on Simulated Datasets with 16 Nodes.

| METRIC | METHOD | LINEAR | LINEAR SPN | NONLINEAR | NONLINEAR SPN |
|--------|--------|--------|------------|-----------|---------------|
| F1 | BaCaDi | $0.18 \pm 0.02$ | $0.22 \pm 0.03$ | $\underline{0.16 \pm 0.03}$ | $0.20 \pm 0.03$ |
| | DECI | $0.09 \pm 0.02$ | $0.11 \pm 0.01$ | $0.08 \pm 0.01$ | $0.08 \pm 0.02$ |
| | VI-DP-DAG | $0.20 \pm 0.04$ | $0.20 \pm 0.03$ | $0.13 \pm 0.00$ | $0.21 \pm 0.06$ |
| | ProDAG | $0.17 \pm 0.01$ | $0.20 \pm 0.02$ | $\underline{0.16 \pm 0.03}$ | $0.23 \pm 0.05$ |
| | ABCDEFG | $\mathbf{0.74 \pm 0.13}$ | $\mathbf{0.49 \pm 0.13}$ | $\mathbf{0.23 \pm 0.31}$ | $\mathbf{0.35 \pm 0.24}$ |
| | ABCDEFG (SPN) | $\underline{0.40 \pm 0.03}$ | $\underline{0.24 \pm 0.06}$ | $0.13 \pm 0.13$ | $\underline{0.30 \pm 0.24}$ |
| SHD | BaCaDi | $108.28 \pm 0.95$ | $106.50 \pm 1.49$ | $109.34 \pm 0.82$ | $107.78 \pm 0.85$ |
| | DECI | $37.27 \pm 0.76$ | $\underline{41.89 \pm 5.36}$ | $36.75 \pm 4.17$ | $41.88 \pm 4.35$ |
| | VI-DP-DAG | $83.88 \pm 4.32$ | $79.48 \pm 1.97$ | $86.51 \pm 5.35$ | $79.78 \pm 4.28$ |
| | ProDAG | $98.24 \pm 1.56$ | $94.79 \pm 1.56$ | $81.16 \pm 0.60$ | $88.00 \pm 2.52$ |
| | ABCDEFG | $\mathbf{12.74 \pm 5.02}$ | $\mathbf{29.25 \pm 3.57}$ | $\mathbf{22.14 \pm 5.44}$ | $\mathbf{27.68 \pm 0.88}$ |
| | ABCDEFG (SPN) | $\underline{34.40 \pm 1.80}$ | $43.11 \pm 2.86$ | $\underline{30.38 \pm 6.76}$ | $\underline{34.31 \pm 3.87}$ |

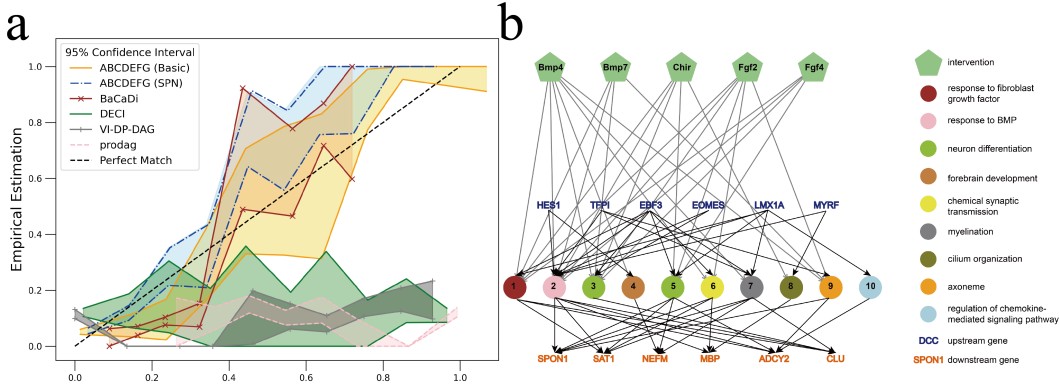

Figure 2: **Posterior calibration plot of Bayesian methods and extended factor graph inferred from growth factor screen. (a)** 95% confidence intervals estimated empirically (colored regions) across the range of posterior edge probabilities for each method. The black dotted line indicates perfect calibration. **(b)** Inferred causal edges among interventions with unknown targets (growth factors; pentagons), factors (circles), and genes (text) are shown. Factor colors indicate gene ontology terms enriched in the upstream (blue) and downstream (orange) genes. Edges from interventions to factors are shown in gray arrows, and edges between genes and factors are shown in black arrows.

intervention combinations during training, then calculated the MSE of reconstructed data on these held-out interventions. ABCDEFG achieved better MSE on the held-out samples (Basic: 0.917, SPN: 0.922) compared with DCDFG (0.957) and SDCD (1.029). Finally, we visualized the causal factor graph learned by ABCDEFG (Fig. 2b).

## 4  CONCLUSION

ABCDEFG fills a key gap in the field by enabling scalable Bayesian causal discovery from interventional data with known or unknown intervention targets. However, we acknowledge several limitations. First, gene regulatory networks often contain cycles, violating the acyclicity assumption. Second, the f-DAG approach could poorly approximate a causal DAG when the true graph is high-rank (or when the number of factors in the f-DAG is too low). Also, our identifiability theorems do not describe the influence of sample size, though we think that our framework provides a promising foundation for future efforts to extend identifiability results into the limited data regime. ABCDEFG opens exciting new opportunities to infer gene regulatory networks and perturbation targets from large-scale cellular perturbation data.

## 5 REPRODUCIBILITY STATEMENT

To ensure reproducibility, we provide anonymous code for ABCDEFG as a supplementary file. Details of the sum-product network are described in Appendix Section A. We also describe the assumptions of our theorems in more detail and provide complete proofs in Appendix Section B. We describe simulated data generation in detail in Appendix Section C, and details of the real data preprocessing are given in Appendix Section D.

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

# A   OVERVIEW OF SUM-PRODUCT NETWORK

Using the generative process we developed for constructing extended factor graphs, we can infer a causal DAG by optimizing a score function with respect to two binary matrices $Y$ and $B$. But what is the best way to do this, given that $Y$ and $B$ are discrete? One possibility is the Gumbel softmax trick [13], often applied due to its simplicity. For $Y$, we can parameterize each $Y_i$ with logits $\theta_i$ and sample $Y_i$ using Gumbel softmax. Similarly we can treat each edge in $B$ as a Bernoulli random variable and sample from Gumbel softmax. However, such a naive approach treats all edges as independent and neglects possible correlation between edges.

A more general approach is to model the joint distribution of edges in $B$ using a sum-product network [20]. Two naive ways to sample a binary vector $\mathbf{b} \in \{0,1\}^d$ are (1) sample from a single categorical distribution over all binary vectors or (2) sample each entry independently from a Bernoulli distribution. The former involves $2^d$ categories, which is impractical for large $d$, while the latter neglects dependency between any two entries and lacks expressiveness. In contrast, SPNs provide an appealing parametric model for $B$ due to their balance between model complexity and expressiveness.

Let $B = [B_1, \ldots, B_d]^T \in \{0,1\}^d$ be a random binary vector. We applied and extended the algorithm by Shih & Ermon [23] to construct an SPN to model the joint distribution of $B$. The construction of an SPN is analogous to building a neural network by sequentially adding layers. Each layer contains one type of computation nodes: (1) input node, (2) product node and (3) sum node and acts as a function of input as shown in Fig. 3a. The SPN starts with singletons $\{b_1\}, \ldots, \{b_d\}$ as an initial partition. Each $b_i$ is passed to two input nodes outputting 0 and 1 respectively. Next, each product layer merges the partitions from the previous layer by creating all combinations of bit sequences for each merge. When the number of sequences from a merge exceeds a threshold, $w$, a sum layer is added to filter out sequences from the previous layer while keeping the same number of partitions. The merge filter process continues until a single partition remains. Thus, the SPN can also be interpreted as a deep mixture model whose trainable parameters are the mixture weights of all sum nodes.

The original algorithm by Shih & Ermon [23] only works when $d$ is a power of two due to recursively halving the partitions, but we extended it to the general case. To do this, we divide $d$ into powers of two based on its binary representation: $d = \sum_{i=0}^{k} b_i \times 2^i$. Next, for each $b_i = 1$, we build an SPN modeling joint PMF of $2^i$ bits. Finally, we apply a product and sum unit to merge the outputs from each SPN together. The number of parameters in an SPN with a maximum width of $w$ for an f-DAG with $m$ factors and $n$ nodes scales as $\Theta\left(\frac{mnw^2}{\log w}\right)$, achieving a balance between model size and model expressiveness.

We further provide a theoretical bound on the space complexity of the SPN-FG model we used for ABCDEFG.

**Notation.** As introduced in section 2.3, an SPN-FG model contains partition variables $Y = \{Y_i : i \in [n]\}$ and connection matrix $B \in \{0,1\}^{n \times m}$ parameterized by sum-product networks (SPN). We use the following notation throughout the derivation.

1. $n$: number of graph nodes.

2. $m$: number of factors.

3. $l$: SPN layer index

4. $p_l$: number of partitions in the $l$-th layer of an SPN

5. $u_l$: number of sum or product nodes in each partition in the $l$-th layer of an SPN.

6. $w$: maximum number of bit sequences from a product node.

7. $s$: Total number of trainable parameters of a single SPN.

8. $S$: Total number of trainable parameters of an SPN-FG model.

We define model complexity as the total number of trainable parameters of an SPN-FG. In our implementation, the joint PMF of either a row or a column of $B$ can be parameterized with a separate SPN. We consider the case of building an SPN for each row of $B$, i.e. each SPN models the joint

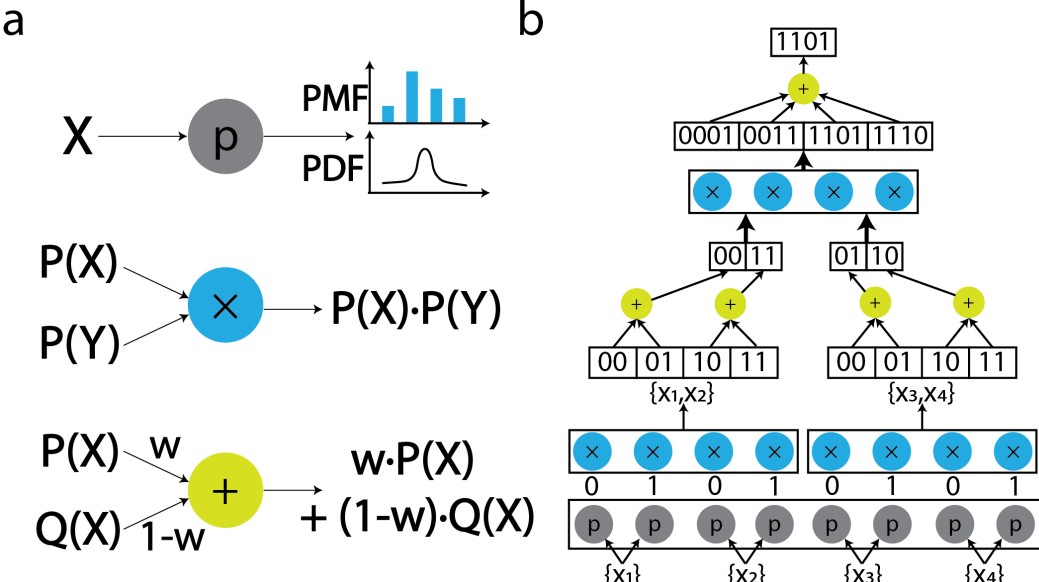

Figure 3: **Illustration of Sum-Product Network (SPN). (a)** Building blocks of an SPN. **Top**: An input node encodes a PMF or PDF given an input value $x$. **Middle**: A product node generates a product of input distributions as the output. **Bottom**: A sum node generates a mixture of input distributions as the output. **(b)** An Example of SPN Architecture. Assume inputs are random bits $x_1 \ldots x_4$. Input nodes generate both 0's and 1's for each bit. Next, a product layer merges $x_1, x_2$ and $x_3, x_4$ by generating all 2-bit sequences for $\{x_1, x_2\}$ and $\{x_3, x_4\}$ respectively. Then, a sum layer downsamples inputs. Finally, a product and a sum layer merge $x_1, \ldots, x_4$ together and output a 4-bit sequence.

distribution of connections between one node and all factors. This results in the following general formula for trainable parameters:

$$S = n(m + 1) + ns \tag{2}$$

The first part $n(m + 1)$ represents $n$ categorical distributions with $m + 1$ categories for modeling $\boldsymbol{Y}$. The second part $ns$ represents $n$ SPNs, each having $s$ parameters and modeling a single row of $\boldsymbol{B}$. Later, we will see that the space complexity stays the same when we choose to parameterize each column of $\boldsymbol{B}$ with an SPN. Notice that $s$ is a function of $m$, $n$ and $w$. Next, we derive bounds of $s$.

**Special Case.** Here, we consider a special case of both $m$ and $w$ being a power of 2. Suppose $m = 2^d$ and $w = 2^k$. This is also the assumption in the original algorithm by Shih & Ermon [23].

We build an SPN by sequentially adding either a product or a sum layer to the network. The algorithm by Shih et al. keeps adding product layers until the Cartesian product of two partitions has a size exceeding the bound $w$. Here, we further assume $w < 2^m$ because if it's not the case, the width bound, $w$, has no effect and the SPN will be equivalent to a categorical distribution over all $2^m$ binary vectors. Once the width of an SPN exceeds $w$, we add sum and product layers alternatingly. Each sum node constraints the partition size $u_l$ to be $w$, while each product node always combines two sets of $w$ sequences into $w^2$ combinations. That is, we have

$$p_l = \begin{cases} \frac{1}{2}p_{l-1} & l \leq L_0 \\ p_{l-1} & l > L_0, l - L_0 \text{ odd (sum)} \\ \frac{1}{2}p_{l-1} & l > L_0, l - L_0 \text{ even (product)} \end{cases} \tag{3}$$

$$u_l = \begin{cases} u_{l-1}^2 & l \leq L_0 \\ w & l > L_0, l - L_0 \text{ odd (sum)} \\ w^2 & l > L_0, l - L_0 \text{ even (product)} \end{cases} \tag{4}$$

Here $L_0 + 1$ is the lowest index of the layer whose partition size exceeds the budget $w$, i.e. $L_0 := \max_l u_l \leq w$. Using Eq. 3-4, we have $u_l = 2^{2^l}$ when $l < L_0$. This implies

$$L_0 := \max\{l : 2^{2^l} \leq 2^k\} \implies L_0 = \lfloor \log_2 k \rfloor. \tag{5}$$

The trainable parameters of our SPN are the mixture weights of sum nodes and in each sum layer, the number of sum nodes equals the number of partitions times number of nodes for each partition. Therefore, the total number of trainable parameters of each SPN equals:

$$s = p_{L_0+1} \cdot u_{L_0+1} + \sum_{l'=1}^{d-L_0-1} p_{L_0+2l'+1} \cdot u_{L_0+2l'+1} \tag{6}$$

$$= 2^{2^{L_0+1}} \frac{m}{2^{L_0+1}} + \sum_{l'=1}^{d-L_0-1} \frac{m}{2^{L_0+1+l'}} w^2 \tag{7}$$

$$= 2^{2^{L_0+1}-L_0-1} m + mw^2 \left( \frac{1}{2^{L_0+1}} - \frac{1}{2^d} \right) \tag{8}$$

From Eq. 5, we have

$$\log_2 k - 1 < L_0 \leq \log_2 k$$

$$\iff \frac{k}{2} < 2^{L_0} \leq k$$

$$\iff \sqrt{w} = 2^{\frac{k}{2}} < 2^{2^{L_0}} \leq 2^k = w. \tag{9}$$

By plugging the upper and lower bound in the above inequality into Eq. 8, we have

$$\frac{mw}{2 \log w} + mw^2 \left( \frac{1}{2 \log w} - \frac{1}{m} \right) < s < mw^2 \left( \frac{2}{\log w} - \frac{1}{m} \right) \tag{10}$$

$$\implies s = \Theta \left( \frac{mw^2}{\log w} \right) \tag{11}$$

$$\implies S = n(m+1) + ns = \Theta \left( \frac{mnw^2}{\log w} \right). \tag{12}$$

When each SPN models a column of $B$ instead of a row, we have $s = \Theta(\frac{nw^2}{\log w})$ and hence,

$$S = n(m+1) + ms = \Theta \left( \frac{mnw^2}{\log w} \right).$$

Finally, we conclude that

$$S = \Theta \left( \frac{mnw^2}{\log w} \right).$$

Now we consider the alternative way of modeling each column of $B$ with an SPN. Then, the total number of parameters becomes

$$S = n(m+1) + ms.$$

Following exactly the same derivation with $m$ replaced with $n$, we have each SPN $s = \Theta(\frac{nw^2}{\log w})$ and the overall $m$ parallel SPNs have a space complexity of $S = \Theta \left( \frac{mnw^2}{\log w} \right)$. Hence, we end up with the same space complexity.

**General Case.** Because the number of nodes or factors cannot always be a power of two, we would like to extend the original algorithm by considering any $m$ and $n$. Again, we first consider building $n$ SPNs, each modeling the joint distribution of $m$ entries in a row of $B$. Our algorithm first decomposes $m$ into its binary representation:

$$m = \sum_{i=0}^{r} b_i 2^i, \ b_i \in \{0, 1\}.$$

We build an SPN for each $2^i$ entries, and finally use a product unit to concatenate them together - this comes at a price of not modeling the full joint distribution of all bits, but results in a convenient implementation and nice properties such as decomposability and smoothness. Denote $s(m)$ as the space complexity of an SPN with $m$ input bits. In the special case above, we assume the width bound $w < 2^m$. When $w \geq 2^m$, the SPN is equivalent to a categorical distribution with a support on all possible binary vectors. Thus, we have

$$
s(2^d) = \begin{cases} 2^{2^{L_0+1}-L_0-1}m + mw^2 \left(\frac{1}{2^{L_0+1}} - \frac{1}{2^d}\right) & w < 2^m \\ 2^m & w \geq 2^m \end{cases}
$$

We assume $2^d \leq m < 2^{d+1}$ and $w = 2^k$. Then, we have

$$
s(m) = \sum_{r=0}^{d} b_r s(2^r) + \mathbf{1}_{\{2^r > k \,\wedge\, r < d\}} \cdot w \tag{13}
$$

Here, $\mathbf{1}_{\{\cdot\}}$ is the indicator function. The right hand side of Eq. 13 has two parts. The first part is just sum of all sub-networks. The second part accounts for the fact that $\forall r < d$, the SPN with $2^r$ nodes already reaches the end with $w^2$ combinations and instead of using a single sum node to select one out of $w^2$ nodes, $w$ sum nodes will select one out of every $w$ inputs, and only $w$ outputs come out of the sum layer. Then, a final layer will select one out of the $w$ outputs to output a single value. On the contrary, the largest SPN with $2^d$ nodes will also reach the last product layer with $w^2$ nodes, but then, by the algorithm, the $w^2$ outputs will go through a final layer with a single sum node to select one out of $w^2$ inputs, so it has $w$ fewer parameters than the other sub-networks with $2^r$ nodes ($r < d$).

Using the previous result, we can get an upper and lower bound of $s(m)$.

$$
\begin{aligned}
s(m) &= \sum_{r=0}^{d} b_r s(2^r) + \mathbf{1}_{\{2^r > k \,\wedge\, r < d\}} \cdot w \\
&\geq s(2^d) \\
&\geq \frac{2^d w}{2 \log_2 w} + 2^d w^2 \left(\frac{1}{2\log_2 w} - \frac{1}{2^d}\right) \\
&\geq \frac{(\frac{m}{2}+1)w}{2\log_2 w} + (\frac{m}{2}+1)w^2 \frac{2}{\log_2 w} - w^2.
\end{aligned} \tag{14}
$$

The upper bound depends on $L_0 = \lfloor \log_2 k \rfloor$ because small networks do not need sum layers to pre-select outputs from product nodes.

$$
\begin{aligned}
s(m) &= \sum_{r=0}^{d} b_r s(2^r) + \mathbf{1}_{\{2^r > k \,\wedge\, r < d\}} \cdot w \\
&\leq \sum_{r=0}^{L_0} 2^{2^r} + \sum_{r=L_0+1}^{d} s\left(2^r\right) + (d - L_0 - 1) \cdot w \\
&\leq \sum_{r=0}^{L_0} 2^{2^r} + \sum_{r=L_0+1}^{d} 2^r w^2 \left(\frac{2}{\log_2 w} - \frac{1}{2^r}\right) + (d - L_0 - 1) \cdot w \\
&\leq 2^{2^{L_0}} \cdot L_0 + \left(2^{d+2} - 2^{L_0+2}\right)\frac{w^2}{\log_2 w} - (d - L_0)\,w^2 + (d - L_0 - 1)w \\
&< w\left[\log_2 \log_2(w)\right] + \frac{(4m - 2\log_2 w)w^2}{\log_2 w} - (\log_2(m) - 1 - \log_2 \log_2(w))\,w^2 \\
&\quad + (\log_2(m) - \log_2 \log_2(w))w \\
&= w\log_2(m) + \frac{(4m - 2\log_2 w)w^2}{\log_2 w} - (\log_2(m) - 1 - \log_2 \log_2(w))\,w^2.
\end{aligned} \tag{15}
$$

Therefore, we have $s(m) = \Theta(\frac{mw^2}{\log w})$ and consequently

$$
S = \Theta\left(\frac{mnw^2}{\log w}\right).
$$

# B  IDENTIFIABILITY OF CAUSAL DAGs BY ABCDEFG

In this section, we will introduce key concepts from existing literature[27; 6; 17] and prove the identifiability of our method. Previously, Yang et al. introduced the concept of $\mathcal{I}$-Markov equivalence as an extension of Markov equivalence. Brouillard et al. proved the identifiability of $\mathcal{I}$-Markov equivalent graphs under score maximization. Later, Lopez et al. provided a sufficient condition for a causal DAG to be unique given its corresponding f-DAG. Here, we extend the theory of causal discovery of DAGs and f-DAGs showing (1) a derivation of variational Bayes approach to causal discovery, (2) identifiability of $\mathcal{I}$-Markov equivalent causal graphs under ELBO maximization and (3) a sufficient and necessary condition for equivalence between $\mathcal{I}$-Markov equivalence of f-DAGs and $\mathcal{I}$-Markov equivalence of their half-squared graphs.

## B.1  THEORETICAL FOUNDATION FOR BAYESIAN CAUSAL DISCOVERY OF FACTOR DAGs

We first introduce concepts about causal discovery and factor DAG as from DCDI Brouillard et al. [6] and DCD-FG [17].

**Definition B.1** (Lopez et al. [17]). Given a set of nodes, $V$, and factors, $F$, a factor directed acyclic graph (f-DAG), denoted as $(V, F, E)$, is a directed acyclic graph $(V \cup F, E)$ where edges $E \subset \{(i,j) : i \in V, j \in F \text{ or } i \in F, j \in V\}$.

An f-DAG is a DAG with two different types of vertices, nodes and factors. All edges connect two vertices of different types. Alternatively, if we represent an f-DAG using an adjacency matrix $\mathbf{A}$, we can use $\mathbf{U}$ and $\mathbf{V}$ to represent node-to-factor and factor-to-node adjacency matricies. Then we have $\mathbf{A} = \mathbf{U} \circ \mathbf{V}$ where $\circ$ denotes the matrix Boolean product. Furthermore, we can condense an f-DAG to a node-only graph as defined below.

**Definition B.2** (Lopez et al. [17]). Given an f-DAG, $D = (V, F, E)$, its half-square node graph is defined as $D^2[V] = (V, \{(i,j) : \exists f \in F, (i,f), (f,g) \in E\})$, and half-square factor graph is defined as $D^2[F] = (F, \{(f,g) : \exists i \in V, (f,i), (i,g) \in D\})$.

A half-square graph essentially keeps all dependency relations between nodes in the original factor graph. The factors can be interpreted as intermediate nodes on the paths between causally-related observations. We also note that the mapping from the set of f-DAGs to half-square graphs is an surjection.

Denote $par(\cdot; D)$ and $chd(\cdot; D)$ as the set of parent and child nodes in any graph $D$.

**Definition B.3.** Let $G = (V, E)$ be any graph, $\forall f \in V$, the set of unique parents and children of $f$ are defined as $P_f(G) := \{i : i \in par(f; G), chd(i; G) = \{f\}\}$ and $C_f(G) := \{j : j \in chd(f; G), par(j; G) = \{f\}\}$.

With the above definition, we define a subset of f-DAGs:

Given a set of causally related random variables $\boldsymbol{X} = \{X_1, \dots, X_n\}$ with a causal graph $G$. A fundamental assumption of a causal DAG underlying $\boldsymbol{X}$ is the Markov property, which leads to a factorization of the joint distribution. Here, we denote $\boldsymbol{\pi_i}$ as the set of all parents of $i$ in $G$.

**Definition B.4** (Brouillard et al. [6]). Let $G = (V, E)$ be a causal DAG with $n$ nodes and $\mathcal{I}^* = \{I_k : k \in [l]\}$ be a set of interventions. We define $\mathcal{M}_{\mathcal{I}^*}(G)$ as the set of joint distributions factorized according to the Markov property, i.e. $\boldsymbol{\mathcal{M}}_{\mathcal{I}^*}(G) := \{\{p^{(k)} : k \in [n^{\mathcal{I}^*}]\} : p^{(k)}(\boldsymbol{X}) = \prod_{i=1}^{n} p^{(k)}(X_i | \boldsymbol{X}_{\boldsymbol{\pi_i}})\}$.

By convention, $I_1 = \emptyset$ represents a pure observational setting.

Based on the definition above, Brouillard et al. [6] defined a type of equivalence relation called $\mathcal{I}$-Markov equivalence relation to describe DAG equivalence under interventions.

**Definition B.5** ($\mathcal{I}$-Markov Equivalence [6]). Two DAGs $G_1$ and $G_2$ are $\mathcal{I}$-Markov equivalence if and only if $\boldsymbol{\mathcal{M}}_{\mathcal{I}}(G_1) = \boldsymbol{\mathcal{M}}_{\mathcal{I}}(G_2)$. We denote by $\mathcal{I}$-MEC$(G)$ as the set of all DAGs which are $\mathcal{I}$-Markov equivalent to $G$.

In the rest of section B, we use the notation $\simeq_{\mathcal{I}}$ to denote $\mathcal{I}$-Markov equivalence relation.

Since we consider the set of f-DAGs, the causal relations between $i$ and $j$ are passed through latent factors. Denote $\boldsymbol{\pi}_i^D$ as the set of parents of a vertex $i$(node or factor) in the graph $D$. Next, we use a continuous random variable $\boldsymbol{Z} = \{Z_1, \ldots, Z_m\}$ to represent the factors. Then, we have a class of joint distributions of $\boldsymbol{X}$ and $\boldsymbol{Z}$ produced by an f-DAG.

**Definition B.6** (Family of Distributions associated with an f-DAG). Let $D = (V, F, E)$ be an f-DAG with $n$ nodes and $m$ factors. Then, $\boldsymbol{\mathcal{M}}_{\mathcal{I}^*}(D)$ is defined as the set of probabilistic models with the following form:

$$
\boldsymbol{\mathcal{M}}_{\mathcal{I}^*}(D) = \left\{ \{p^{(k)}(\boldsymbol{X}, \boldsymbol{Z}) : k \in [n^{\mathcal{I}^*}]\} : p^{(k)}(\boldsymbol{X}, \boldsymbol{Z}) = \prod_{i=1}^n p^{(k)}(X_i | \boldsymbol{Z}_{\boldsymbol{\pi}_i^D}) \prod_{j=1}^m p^{(k)}(Z_j | \boldsymbol{X}_{\boldsymbol{\pi}_j^D}) \right\}
$$
(16)

where $p^{(k)}(X_i | \boldsymbol{X}_{\boldsymbol{\pi}_i^D}) \neq p^{(1)}(X_i | \boldsymbol{X}_{\boldsymbol{\pi}_i^D})$ if and only if $i \in I_k$ and $p^{(k)}(Z_j | \boldsymbol{X}_{\boldsymbol{\pi}_j^D}) \neq p^{(1)}(Z_j | \boldsymbol{X}_{\boldsymbol{\pi}_j^D})$ if and only if $j \in I_k$.

The above definition assumes knowledge of the intervention targets. When interventions are unknown, we are able to extend f-DAGs in a similar way to the $\mathcal{I}$-DAG introduced by Yang et al.[27]. We first mention the concept of $\mathcal{I}$-DAG and then extend it to f-DAGs.

**Definition B.7** (Yang et al. [27]). Let $G = (V, E)$ be a DAG and $\mathcal{I} = \{I_1, \ldots, I_{n^{\mathcal{I}}}\}$ be a set of interventions with $I_k \subseteq V, \forall k$. An interventional DAG ($\mathcal{I}$-DAG) is defined as an augmented graph

$$
G^{\mathcal{I}} = (V \cup \Xi, E \cup E^{\mathcal{I}}),
$$

where $\Xi := \{\xi_k : k \in [n^{\mathcal{I}}]\}$ is a set of intervention nodes representing $I_1, \ldots, I_k$ and $E^{\mathcal{I}} \subseteq \{(\xi_k, i) : i \in I_k, k \in [n^{\mathcal{I}}]\}$ is a set of edges from interventions to targets.

**Definition B.8** (Extended f-DAG). Let $D = (V, F, E)$ be an f-DAG and $\mathcal{I} = \{I_1, \ldots, I_{n^{\mathcal{I}}}\}$ be a set of interventions. Let $\Xi = \{\xi_k, k \in [n^{\mathcal{I}}]\}$ be $n^{\mathcal{I}}$ nodes corresponding to the $l$ interventions. An extended f-DAG is defined as an f-DAG $D^{\mathcal{I}} = (V \cup \Xi, F, E \cup E^I)$ where $E^{\mathcal{I}} \subseteq \{(\xi_k, f) : f \in F\}$, i.e. set of edges from intervention nodes to factors.

An extended f-DAG is obtained by adding intervention nodes to an f-DAG. Here, we also have low-rank assumption that interventions causally affects downstream nodes via a small number of factors. Put in a matrix form, the adjacency matrix of an extended f-DAG has a low-rank Boolean matrix factorization as

$$
\mathbf{A}^{\mathcal{I}} = \begin{bmatrix} \mathbf{U} \\ \mathbf{W} \end{bmatrix} \circ [\mathbf{V} \; \mathbf{0}_{m \times n^{\mathcal{I}}}],
$$

where $\mathbf{W} \in \mathbb{R}^{n^{\mathcal{I}} \times m}$ is an adjacency matrix representing edges from intervention nodes to factors.

Given the definition of $\boldsymbol{\mathcal{M}}_{\mathcal{I}^*}(D)$ and $\mathcal{I}$-Markov equivalence, we can further define $\mathcal{I}$-Markov equivalence relation between f-DAGs.

**Definition B.9** ($\mathcal{I}$-Markov Equivalence Class of f-DAGs). Given a set of interventions, $\mathcal{I}$, two f-DAGs $D_1$ and $D_2$ are $\mathcal{I}$-Markov equivalent if $\boldsymbol{\mathcal{M}}_{\mathcal{I}}(D_1) = \boldsymbol{\mathcal{M}}_{\mathcal{I}}(D_2)$.

The concept of $\boldsymbol{\mathcal{M}}_{\mathcal{I}}(D)$ and $\mathcal{I}$-Markov equivalence for f-DAGs are just the same as those for DAGs except for classifying vertices into nodes and factors.

The following theorem regarding the concept of $\mathcal{I}$-DAG connects statistical independence to graph structures.

**Theorem B.10** (Yang et al. [27]). *Two DAGs $G_1$ and $G_2$ belong to the same $\mathcal{I}$-Markov Equivalence Class ($\mathcal{I}$-MEC) if and only if their $\mathcal{I}$-DAGs have the same skeleton and v-structures.*

Since f-DAGs are one type of DAG, we easily obtain the following corollary.

**Corollary B.11.** *Two f-DAGs $D_1$ and $D_2$ belong to the same $\mathcal{I}$-MEC if and only if their extended f-DAGs have the same skeleton and v-structures.*

*Proof.* Suppose $D_1$ and $D_2$ have $n$ nodes and $m$ factors. Let $G_1$ and $G_2$ be two DAGs obtained by removing the labeling of node or factor in $D_1$ and $D_2$. We still keep the bijection between vertices

and random variables $\boldsymbol{X} = \{X_i : i \in [n]\}$ and $\boldsymbol{Z} = \{Z_j : j \in [m]\}$. Then, $G_1$ and $G_2$ are $\mathcal{I}$-Markov equivalent by Theorem B.10. By the definition of $\mathcal{I}$-Markov equivalence, we have

$$\mathcal{M}_{\mathcal{I}}(G_1) = \mathcal{M}_{\mathcal{I}}(G_2) \implies p_1^{(k)}(\boldsymbol{X}, \boldsymbol{Z}) = p_2^{(k)}(\boldsymbol{X}, \boldsymbol{Z}) \forall k$$
$$\implies \mathcal{M}_{\mathcal{I}}(D_1) = \mathcal{M}_{\mathcal{I}}(D_2) \implies D_1 \in \mathcal{I}\text{-MEC}\,(D_2)$$

The first implication comes from the Markov property (d-separation in graphs implies conditional independence). The second implication comes from definition of extended f-DAGs. The last implication comes from the definition of $\mathcal{I}$-Markov equivalence class of f-DAGs. $\qquad\square$

In reality, we can use a single encoder function to get $Z_j \sim p(f_{enc}(\mathbf{U_j} \odot \boldsymbol{X}; \boldsymbol{\Theta}))$ and $X_i \sim p(f_{dec}(\mathbf{V_i} \odot \boldsymbol{X}; \boldsymbol{\Phi}))$ to represent the conditional distribution $p^{(k)}(X_i | \boldsymbol{Z}_{\boldsymbol{\pi}_i^P})$ and $p^{(k)}(Z_j | \boldsymbol{X}_{\boldsymbol{\pi}_j^P})$. Thus, we define a second set of joint distributions representing our model capacity.

**Definition B.12** (Family of Parametric Distributions associated with an f-DAG). Let $D = (V, F, E)$ be an f-DAG with $n$ nodes and $m$ factors. Consider two parametric functions $f_{enc} : \mathbb{R}^n \to \mathbb{R}^m$, parameterized by $\boldsymbol{\Theta} \in \Omega(\boldsymbol{\Theta})$ and $f_{dec} : \mathbb{R}^m \to \mathbb{R}^n$, parameterized by $\boldsymbol{\Phi} \in \Omega(\boldsymbol{\Phi})$. In addition, let $\mathbf{U}$ and $\mathbf{V}$ be node-to-factor and factor-to-node matrices of an f-DAG $D$. Then, $\boldsymbol{\mathcal{F}}_{\mathcal{I}^*}(D)$ is defined as the set of probabilistic models with the following form:

$$\boldsymbol{\mathcal{F}}_{\mathcal{I}^*}(D) = \left\{ \{f^{(k)}(\boldsymbol{X}, \boldsymbol{Z}) : k \in [n^{\mathcal{I}^*}]\} : f^{(k)}(\boldsymbol{X}, \boldsymbol{Z}) = \prod_{i=1}^{n} f^{(k)}(X_i | \boldsymbol{Z}_{\boldsymbol{\pi}_i^P}) \prod_{j=1}^{m} f^{(k)}(Z_j | \boldsymbol{X}_{\boldsymbol{\pi}_j^P}) \right\}, \tag{17}$$

where $f^{(k)}(Z_j | \boldsymbol{X}_{\boldsymbol{\pi}_j^P}) = p(f_{enc}(\mathbf{U_j} \odot \boldsymbol{X}))$, $f^{(k)}(X_i | \boldsymbol{Z}_{\boldsymbol{\pi}_i^P}) = p(f_{enc}(\mathbf{V_i} \odot \boldsymbol{Z}))$, $f^{(k)}(X_i | \boldsymbol{Z}_{\boldsymbol{\pi}_i^P}) \neq f^{(1)}(X_i | \boldsymbol{Z}_{\boldsymbol{\pi}_i^P})$ if and only if $i \in I_k$ and $f^{(k)}(Z_j | \boldsymbol{X}_{\boldsymbol{\pi}_j^P}) \neq f^{(1)}(Z_j | \boldsymbol{X}_{\boldsymbol{\pi}_j^P})$ if and only if $j \in I_k$.

## B.2 DERIVATION OF BAYESIAN FRAMEWORK FOR DIFFERENTIABLE CAUSAL DISCOVERY

We present a Bayesian framework for differentiable causal discovery and show that it reduces to score maximization under a uniform prior over the space of DAGs.

Consider a set of causally related random variables $\boldsymbol{X} = \{X_i : i \in [n]\}$ and a random intervention set $I^* \subseteq [n]$. First, we assume the observations are generated from a single causal graph $G^*$ via a generative model $p(\boldsymbol{X} | G^*, I^*)$. We assume each intervention either removes edges towards targets (hard) or keeps the same graph structure (soft). Thus, the generative model becomes $p(\boldsymbol{X} | G^*, I^*)$ under different interventions. When $I^*$ is known, we can obtain a MAP estimate of $G$:

$$\hat{G} = \arg\max_{G \in \boldsymbol{\mathcal{G}}} p(G | \boldsymbol{X}, I^*). \tag{18}$$

In order to convert this optimization problem to a differentiable one, we consider a variational distribution $q(G)$ and optimize a KL divergence instead:

$$\hat{G} = \arg\max_{G} q^*(G) \text{ where } q^*(G) = \arg\min_{q(G)} KL(q(G) || p(G | \boldsymbol{X}, I^*)). \tag{19}$$

Because we have control over $q(G)$, finding its mode will be easy. Directly optimizing $KL(q(G) || p(G | \boldsymbol{X}))$ suffers from the intractability problem since $p(G | \boldsymbol{X}, I^*) = \frac{p(\boldsymbol{X} | G, I^*) p(G | I^*)}{\sum_{G'} p(\boldsymbol{X} | G', I^*) p(G' | I^*)}$ and the space of DAGs is super-exponential in the number of nodes. Thus,

we can derive an alternative objective in the following form:

$$KL(q(G)||p(G|\boldsymbol{X}, I^*))$$

$$= \mathbb{E}_{p(\boldsymbol{X}, I^*)}\left[\mathbb{E}_{q(G)}\left[\log\frac{q(G)}{p(G|\boldsymbol{X}, I^*)}\right]\right]$$

$$= \mathbb{E}_{p(\boldsymbol{X}, I^*)}\left[\mathbb{E}_{q(G)}\left[\log\frac{q(G)p(\boldsymbol{X}|I^*)}{p(\boldsymbol{X}|G, I^*)p(G|I^*)}\right]\right]$$

$$= \mathbb{E}_{p(\boldsymbol{X}, I^*)}\left[\log p(\boldsymbol{X}|I^*) - \mathbb{E}_{q(G)}\left[\log p(\boldsymbol{X}|G, I^*)\right] + KL(q(G)||p(G|I^*))\right]$$

$$\implies \min_{q(G)} KL(q(G) \,||\, p(G|\boldsymbol{X}, I^*))$$

$$= \max_{q(G)} \mathbb{E}_{p(\boldsymbol{X}, I^*)}\left[\mathbb{E}_{q(G)}\left[\log p(\boldsymbol{X}|G, I^*)\right] - KL(q(G)||p(G|I^*))\right]$$

$$= \max_{q(G)} ELBO(G) \tag{20}$$

In reality, $p(\boldsymbol{X}, I^*)$ is replaced with an empirical distribution from any dataset. For $I^*$, we can conduct additional experiments by perturbing some nodes $I_k$ in the $k$-th experiment. For the empirical data distribution, we assume the data samples are generated from $p(\boldsymbol{X}|G^*)$ instead of $p(\boldsymbol{X})$. The data samples are not drawn from the marginal over $\boldsymbol{X}$ because we assume a single causal graph $G^*$ underlying the data generative process. We use parametric models $f^{(k)}(X_i|\boldsymbol{Z}_{\boldsymbol{\pi}_i^D}; \boldsymbol{\Phi})$ and $f^{(k)}(Z_j|\boldsymbol{X}_{\boldsymbol{\pi}_j^D}; \boldsymbol{\Theta})$ for distributional fitting and $q(G; \boldsymbol{\Lambda})$ for graph fitting. In addition, we need to add an L1 regularization on $G$ to account for the sparsity constraint. Now the optimization problem becomes:

$$\sup_{\boldsymbol{\Lambda}, \boldsymbol{\Phi}} \sum_{k=1}^{n^{\mathcal{I}}} \mathbb{E}_{p^{(k)}(\boldsymbol{X}|G^*)}\left[\mathbb{E}_{q(G; \boldsymbol{\Lambda})}\left[\log f^{(k)}(\boldsymbol{X}|G; \boldsymbol{\Phi})\right] - KL(q^{(k)}(G; \boldsymbol{\Lambda})||p^{(k)}(G))\right] - \lambda|G| \tag{21}$$

The objective function is similar to the one proposed in the VAE paper [14] except that we have a latent space of DAGs instead of a low-dimensional latent embedding. In addition, we assume interventions change neither the prior graph distribution nor our variational posterior. The objective can be extended to that of a $\beta$-VAE:

$$\sup_{\boldsymbol{\Phi}, \boldsymbol{\Lambda}} \sum_{k=1}^{n^{\mathcal{I}}} \mathbb{E}_{p^{(k)}(\boldsymbol{X}|G^*)}\left[\mathbb{E}_{q(G; \boldsymbol{\Lambda})}\left[\log f^{(k)}(\boldsymbol{X}|G; \boldsymbol{\Phi})\right]\right] - \beta KL(q(G; \boldsymbol{\Lambda})||p(G)) - \lambda|G|$$

$$= \sup_{\boldsymbol{\Lambda}} \mathbb{E}_{q(G; \boldsymbol{\Lambda})}\left[\sup_{\boldsymbol{\Phi}} \sum_{k=1}^{n^{\mathcal{I}}} \mathbb{E}_{p^{(k)}(\boldsymbol{X}|G^*)}\left[\log f^{(k)}(\boldsymbol{X}|G; \boldsymbol{\Phi})\right] - \lambda|G|\right] - \beta KL(q(G; \boldsymbol{\Lambda})||p(G)) \tag{22}$$

Notice that the score function is under the expectation of $q(G; \boldsymbol{\Lambda})$. If we set $\beta = 0$ and $q(G; \boldsymbol{\Lambda}) = \delta(G)$, the Dirac delta function, the optimization problem becomes exactly the same as a score maximization problem as presented in previous score-based methods. The constraint on $q(G; \boldsymbol{\Lambda})$ ensures that $q(G; \boldsymbol{\Lambda})$ does not deviate from the prior arbitrarily. Next, we will prove the identifiability of this Bayesian framework.

**Theorem B.13** (Brouillard et al. [6]). *Let $\boldsymbol{X} = \{X_1, \ldots, X_n\}$ be a set of causally related random variables with a causal DAG $G^* = (V, E)$ and $\mathcal{I}^* = \{I_k : k \in [n^{\mathcal{I}^*}]\}$ be a set of interventions with $I_1 = \emptyset$. Assume the following:*

1. *The set of distributions from our parametric models contains the ground truth interventional distributions: $\{p^{(k)}(\boldsymbol{X}) : k \in [n^{\mathcal{I}^*}]\} \in \mathcal{F}_{\mathcal{I}^*}(G^*)$ where $\mathcal{F}_{\mathcal{I}^*}(G^*) = \{\{f^{(k)}(\boldsymbol{X}|G^*; \boldsymbol{\Phi})\} : \boldsymbol{\Phi} \in \Omega(\boldsymbol{\Phi})\}$.*

2. *Denote $\perp\!\!\!\perp_{G^*}$ as the d-separation relation in $G^*$. $\mathcal{I}$-faithfulness contains the following two conditions.*

   (a) *For any disjoint set $A, B, C \subset V$, $\boldsymbol{X_A} \perp\!\!\!\perp \boldsymbol{X_B}|\boldsymbol{X_C} \implies A \perp\!\!\!\perp_{G^*} B|C$*

   (b) *For any disjoint sets $A, C \subset V$ and $k \in [n^{\mathcal{I}^*}]$, $p^{(k)}(\boldsymbol{X_A}|\boldsymbol{X_C}) = p^{(1)}(\boldsymbol{X_A}|\boldsymbol{X_C}) \implies A \perp\!\!\!\perp_{G^* \mathcal{I}^*} \xi_k|C$*

3. $\forall G, I, \mathbf{\Phi}, f^{(k)}(\boldsymbol{X}|G, I; \mathbf{\Phi}) > 0$.

4. $\forall k \in [n^{\mathcal{I}^*}], \left| \mathbb{E}_{p^{(k)}(\boldsymbol{X})} \left[ \log p^{(k)}(\boldsymbol{X}) \right] \right| < +\infty$.

*Define the score function as*

$$S_{\mathcal{I}^*}(G) = \sup_{\mathbf{\Phi}} \sum_{k=1}^{n^{\mathcal{I}^*}} \mathbb{E}_{p^{(k)}(\boldsymbol{X})} \left[ \log f^{(k)}(\boldsymbol{X}|G; \mathbf{\Phi}) \right] - \lambda |G|$$

*Then, with a small enough $\lambda > 0$, we have $S_{\mathcal{I}^*}(G^*) > S_{\mathcal{I}^*}(G)$.*

The previous theorem claims optimality of the score function when the causal DAG is treated as a deterministic object. Next, we give a probabilistic view of this optimality. First, we define the Bayesian score function as follows.

**Definition B.14** (Bayesian Score Function). Let $\boldsymbol{X} = \{X_1, \dots, X_n\}$ be a set of causally related random variables with a causal DAG $G^*$ and $\mathcal{I}^* = \{I_k : k \in [n^{\mathcal{I}}]\}$ be a set of interventions with $I_1 = \emptyset$. Let $p(G)$ be a prior over DAGs and $q(G; \mathbf{\Lambda})$ be a variational distribution. The Bayesian score function, $\mathcal{L}(q(G; \mathbf{\Lambda}))$ is defined as

$$\mathcal{L}(q(G; \mathbf{\Lambda})) = \mathbb{E}_{q(G; \mathbf{\Lambda})} \left[ S_{\mathcal{I}^*}(G) \right] - \beta KL(q(G; \mathbf{\Lambda}) || p(G))$$

where $S_{\mathcal{I}^*}(G)$ is the score function defined in Theorem B.13.

**Theorem B.15** (Identifiability via ELBO maximization). *Let $\boldsymbol{X} = \{X_1, \dots, X_n\}$ be a set of causally related random variables with a causal DAG $G^*$ and $\mathcal{I}^* = \{I_k : k \in [n^{\mathcal{I}}]\}$ be a set of interventions with $I_1 = \emptyset$. Let $\mathcal{G}$ be a subset of all causal DAGs and $q^*(G)$ be an optimal graph distribution from the optimization problem:*

$$\sup_{q(G; \mathbf{\Lambda}): supp(q) \subseteq \mathcal{G}} \mathcal{L}(q(G; \mathbf{\Lambda})),$$

*where*

$$\mathcal{L}(q(G; \mathbf{\Lambda})) = \mathbb{E}_{q(G; \mathbf{\Lambda})} \left[ S_{\mathcal{I}^*}(G) \right] - \beta KL(q(G; \mathbf{\Lambda}) || p(G)),$$

$$S_{\mathcal{I}^*}(G) = \sup_{\mathbf{\Phi}} \sum_{k=1}^{n^{\mathcal{I}}} \mathbb{E}_{p^{(k)}(\boldsymbol{X})} \left[ \log f^{(k)}(\boldsymbol{X}|G; \mathbf{\Phi}) \right] - \lambda |G|.$$

*If $G^* \in \mathcal{G}$, then, under the same assumptions as those in Theorem B.13, for small enough $\beta > 0$ and small enough $\lambda > 0$, $\hat{G} = \arg\max_G q^*(G)$ is $\mathcal{I}^*$-Markov equivalent to $G^*$.*

*Proof.* We prove this theorem by contradiction. Suppose $\exists \hat{G} = \arg\max_G q^*(G)$ such that $\hat{G} \not\simeq_{\mathcal{I}^*} G^*$.

Consider another PMF $q'(G)$ which has the same support and same mass as $q^*(G)$ except for $q'(G^*) - q^*(G^*) = \epsilon > 0$ and consequently, $q'(\hat{G}) - q^*(\hat{G}) = -\epsilon < 0$. Because $q^*(\hat{G}) > 0$, such $\epsilon$ exists. By the definition of $q^*$, $\mathcal{L}(q^*) \geq \mathcal{L}(q')$. Then, we have

$$\mathcal{L}(q') - \mathcal{L}(q^*)$$

$$= \left[ \mathbb{E}_{q'(G)} \left[ S_{\mathcal{I}^*}(G) \right] - \beta KL(q'(G) || p(G)) \right] - \left[ \mathbb{E}_{q^*(G)} \left[ S_{\mathcal{I}^*}(G) \right] - \beta KL(q^*(G) || p(G)) \right]$$

$$= \sum_{G \in \mathcal{G}} (q'(G) - q^*(G)) S_{\mathcal{I}^*}(G) + \beta \left[ KL(q^*(G) || p(G)) - KL(q'(G) || p(G)) \right]$$

$$= \epsilon \left( S_{\mathcal{I}^*}(G^*) - S_{\mathcal{I}^*}(\hat{G}) \right) + \beta \left[ KL(q^*(G) || p(G)) - KL(q'(G) || p(G)) \right]. \tag{23}$$

By Theorem B.13, $\exists \lambda > 0$ such that $S_{\mathcal{I}^*}(G^*) > S_{\mathcal{I}^*}(G), \forall G \not\simeq_{\mathcal{I}^*} G^*$. Therefore, $S_{\mathcal{I}^*}(G^*) - S_{\mathcal{I}^*}(\hat{G}) = \Delta > 0$. If $\sum_{k=1}^{n^{\mathcal{I}^*}} \left[ KL(q^*(G) || p(G)) - KL(q'(G) || p(G)) \right] \geq 0$, we already have $\mathcal{L}(q') > \mathcal{L}(q^*)$. Otherwise, we can pick

$$0 < \beta < \frac{\epsilon \Delta}{KL(q'(G) || p(G) - KL(q^*(G) || p(G)))}$$

and $\mathcal{L}(q') > \mathcal{L}(q^*)$. Both cases contradict the fact that $\mathcal{L}(q^*) \geq \mathcal{L}(q')$. Therefore, we conclude that $G^*$ must be a mode of $q$. $\qquad\square$

Notice that we add a constraint on the support of $q(G; \Lambda)$ to account for cases when we have prior knowledge about the DAG and only need to search over a subset. As discussed below, this applies when the true causal DAG is a half-square graph of an f-DAG. If we set $\mathcal{G}$ to the set of all DAGs, the constraint will be removed.

ABCDEFG aims at optimizing $KL(q(D^2[V])||p(G|\boldsymbol{X}))$ with respect to a distribution on f-DAGs instead of DAGs. As long as the adjacency matrix of the true causal DAG can be factorized as a Boolean product of a node-to-factor and factor-to-node matrices, optimization over f-DAGs guarantees identifiability of the true causal DAG, as a half-square graph of an optimal f-DAG.

## B.3 EXTENSION TO UNKNOWN INTERVENTION TARGETS

**Bayesian framework for interventional causal DAG discovery.** The Bayesian framework can be further extended to unknown intervention targets. Consider a set of causally related random variables $\boldsymbol{X} = \{X_i : i \in [n]\}$ and a random intervention set $I^* \subseteq [n]$. With the same assumptions as the Bayesian framework in section B.2, our goal is to identify the true $\mathcal{I}$-DAG, $(G^*)^{I^*}$. Following a similar argument, we can convert MAP estimation into a continuous optimization problem:

$$\hat{G}^{\mathcal{I}} = \arg\max_{G^{\mathcal{I}} \in \mathcal{G}^{\mathcal{I}}} q^*\left(G^{\mathcal{I}}\right) \text{ where } q^*\left(G^{\mathcal{I}}\right) = \arg\min_{q(G^{\mathcal{I}})} KL\left(q\left(G^{\mathcal{I}}\right) \mid\mid p(G^{\mathcal{I}}|\boldsymbol{X})\right)$$

Alternatively, we can optimize the ELBO as follows:

$$KL(q(G^{\mathcal{I}})||p(G^{\mathcal{I}}|\boldsymbol{X}))$$

$$= \mathbb{E}_{p(\boldsymbol{X})}\left[\mathbb{E}_{q(G^{\mathcal{I}})}\left[\log \frac{q(G^{\mathcal{I}})}{p(G^{\mathcal{I}}|\boldsymbol{X})}\right]\right]$$

$$= \mathbb{E}_{p(\boldsymbol{X})}\left[\mathbb{E}_{q(G^{\mathcal{I}})}\left[\log \frac{q(G^{\mathcal{I}})p(\boldsymbol{X})}{p(\boldsymbol{X}|G^{\mathcal{I}})p(G^{\mathcal{I}})}\right]\right]$$

$$= \mathbb{E}_{p(\boldsymbol{X})}\left[\log p(\boldsymbol{X}) - \mathbb{E}_{q(G^{\mathcal{I}})}\left[\log p(\boldsymbol{X}|G^{\mathcal{I}}\right] + KL(q(G^{\mathcal{I}})||p(G^{\mathcal{I}})]\right] \quad (24)$$

$$\implies \min_{q(G^{\mathcal{I}})} KL(q(G^{\mathcal{I}})||p(G^{\mathcal{I}}|\boldsymbol{X}))$$

$$= \max_{q(G^{\mathcal{I}})} \mathbb{E}_{q(G^{\mathcal{I}})}\left[\log p(\boldsymbol{X}|G^{\mathcal{I}})\right] - KL(q(G^{\mathcal{I}})||p(G^{\mathcal{I}}))$$

$$= \max_{q(G^{\mathcal{I}})} ELBO(G^{\mathcal{I}}) \quad (25)$$

The optimization problem is exactly the same as the one in section B.2, except that we consider a distribution over an $\mathcal{I}$-DAG instead of a DAG. Similar to the derivation of Theorem. B.15, we first present a theorem from Brouillard et al. [6].

**Theorem B.16** (Brouillard et al. [6]). *Let $\boldsymbol{X} = \{X_1, \ldots, X_n\}$ be a set of causally related random variables with a causal DAG $G^* = (V, E)$ and $\mathcal{I}^* = \{I_k : k \in [n^{\mathcal{I}}]\}$ be a set of interventions with $I_1 = \emptyset$. Define the score function as*

$$S(G, \mathcal{I}) = \sup_{\boldsymbol{\Phi}} \sum_{k=1}^{n^{\mathcal{I}}} \mathbb{E}_{p^{(k)}(\boldsymbol{X})}\left[\log f^{(k)}(\boldsymbol{X}|G, \mathcal{I}; \boldsymbol{\Phi})\right] - \lambda|G| - \lambda_R|\mathcal{I}|.$$

*Then, under the same assumptions from Theorem B.13 and with a small enough $\lambda > 0$ and $\lambda_R > 0$, we have $S(G^*, \mathcal{I}^*) > S(G, \mathcal{I})$ for any $G \not\simeq_{\mathcal{I}^*} G^*$ or $\mathcal{I} \neq \mathcal{I}^*$.*

In the implementation, the unknown interventions are parameterized by a binary matrix $R^{\mathcal{I}} \in \{0, 1\}^{n^{\mathcal{I}} \times n}$ where $R^{\mathcal{I}}_{kj} = 1$ if and only if $j \in I_k$. Next, we prove identifiability of our Bayesian framework.

**Theorem B.17** (Identifiability for untargeted interventions via ELBO maximization). *Let (1) $\boldsymbol{X} = \{X_1, \ldots, X_n\}$ be a set of causally related random variables with a causal DAG $G^*$, (2) $\mathcal{I}^* = \{I_k : k \in [n^{\mathcal{I}}]\}$ be a set of unobserved interventions with $I_1 = \emptyset$ and (3) $\mathcal{G}^{\mathcal{I}}$ be the set of all $\mathcal{I}$-DAGs with $n^{\mathcal{I}^*}$ interventions. $\forall G^{\mathcal{I}} \in \mathcal{G}^{\mathcal{I}}$, define $R$ as the adjacency matrix of intervention-to-node graph. Let $q^*(G^{\mathcal{I}})$ be an optimal graph distribution from the optimization problem:*

$$\sup_{q(G^{\mathcal{I}};\boldsymbol{\Lambda}):supp(q)\subseteq \mathcal{G}^{\mathcal{I}}} \mathcal{L}(q(G^{\mathcal{I}};\boldsymbol{\Lambda}))$$

*where*

$$\mathcal{L}(q(G^{\mathcal{I}};\mathbf{\Lambda})) = \mathbb{E}_{q(G;\mathbf{\Lambda})}\left[S(G,\mathcal{I})\right] - \beta KL(q(G^{\mathcal{I}};\mathbf{\Lambda})||p(G^{\mathcal{I}})),$$

$$S(G,\mathcal{I}) = \sup_{\mathbf{\Phi}} \sum_{k=1}^{n^{\mathcal{I}}} \mathbb{E}_{p^{(k)}(\mathbf{X})}\left[\log f^{(k)}(\mathbf{X}|G^{\mathcal{I}};\mathbf{\Phi})\right] - \lambda|G| - \lambda_R|\mathcal{I}|.$$

*If $G^* \in \mathcal{G}$, then, under the same assumptions as those in Theorem B.13, for small enough $\beta > 0$ and small enough $\lambda > 0, \lambda_R > 0$, for any $\hat{G}^{\hat{\mathcal{I}}} = \arg\max_{G^{\mathcal{I}}} q^*(G^{\mathcal{I}})$, $\hat{G} \simeq_{\mathcal{I}^*} G^*$ and $\hat{\mathcal{I}} = \mathcal{I}^*$.*

*Proof.* The proof uses a similar technique as in proof of Theorem B.15.

We prove this theorem by contradiction. Suppose $\exists \hat{G}^{\hat{I}} = \arg\max_G q^*(G^{\mathcal{I}})$ such that $\hat{G} \not\simeq_{\mathcal{I}^*} G^*$ or $\hat{\mathcal{I}} \neq \mathcal{I}^*$.

Consider another PMF $q'(G^{\mathcal{I}})$ which has the same support and same mass as $q^*(G^{\mathcal{I}})$ except for $q'((G^*)^{\mathcal{I}^*}) - q^*((G^*)^{\mathcal{I}^*}) = q^*(\hat{G}^{\hat{\mathcal{I}}}) - q'(\hat{G}^{\hat{\mathcal{I}}}) = \epsilon > 0$. Because $q^*(\hat{G}^{\hat{\mathcal{I}}}) > 0$, such $\epsilon$ exists. By the definition of $q^*$, $\mathcal{L}(q^*) \geq \mathcal{L}(q')$. Then, we have

$$\mathcal{L}(q') - \mathcal{L}(q^*)$$
$$= \left[\mathbb{E}_{q'(G^{\mathcal{I}})}[S(G,\mathcal{I})] - \beta KL(q'(G^{\mathcal{I}})||p(G^{\mathcal{I}}))\right] - \left[\mathbb{E}_{q^*(G^{\mathcal{I}})}[S(G,\mathcal{I})] - \beta KL(q^*(G^{\mathcal{I}})||p(G^{\mathcal{I}}))\right]$$
$$= \sum_{G \in \mathcal{G}^{\mathcal{I}}} (q'(G) - q^*(G))S(G,\mathcal{I}) + \beta\left[KL(q^*(G^{\mathcal{I}})||p(G^{\mathcal{I}})) - KL(q'(G^{\mathcal{I}})||p(G^{\mathcal{I}}))\right]$$
$$= \epsilon\left(S(G^*,\mathcal{I}^*) - S(\hat{G},\hat{\mathcal{I}})\right) + \beta\left[KL(q^*(G^{\mathcal{I}})||p(G^{\mathcal{I}})) - KL(q'(G^{\mathcal{I}})||p(G^{\mathcal{I}}))\right] \tag{26}$$

By Theorem B.16, $\exists \lambda > 0, \lambda_R > 0$ such that $S(G^*,\mathcal{I}^*) > S(\hat{G},\hat{\mathcal{I}}), \forall G \not\simeq_{\mathcal{I}^*} G^*$ or $\mathcal{I} \neq \mathcal{I}^*$. Therefore, $S(G^*,\mathcal{I}^*) - S(\hat{G},\hat{\mathcal{I}}) = \Delta > 0$. If $KL(q'(G,\mathcal{I})||p(G,\mathcal{I})) - KL(q^*(G,\mathcal{I})||p(G,\mathcal{I})) \geq 0$, we already have $\mathcal{L}(q') > \mathcal{L}(q^*)$. Otherwise, we can pick

$$0 < \beta < \frac{\epsilon\Delta}{KL(q'(G,\mathcal{I})||p(G,\mathcal{I})) - KL(q^*(G,\mathcal{I})||p(G,\mathcal{I}))}$$

and $\mathcal{L}(q') > \mathcal{L}(q^*)$. Both cases contradict the fact that $\mathcal{L}(q^*) \geq \mathcal{L}(q')$. Therefore, we conclude that both $\hat{G} \simeq_{\mathcal{I}^*} G^*$ and $\hat{\mathcal{I}} = \mathcal{I}^*$ □

Based on the above results, the proposed model architecture is presented in Fig. 4

## C SUPPLEMENTARY RESULTS

### C.1 RESULTS ON TOY AND EXTENDED DATASETS

We benchmarked existing methods on simulated data using both SPN-FG and previous f-DAG simulation method from Lopez et al. [17]. In a preliminary study, we tested all methods on simple toy datasets simulated with 16 nodes and 2 factors (Table 5). We changed the sparsity penalty in ENCO but it produced mainly zero adjacency matrix except for one datast with 0.13 F1 score. Hence, we report zero F1 scores here as a placeholder. Then we extend our experiment to 200 and 500 nodes with nonlinear intervention, to evaluate the performance on larger graph (Table 6). Note that ENCO and DCDI were too slow and/or required too much memory on larger graphs, so we omitted them from this comparison. We also evaluate our methods on denser graphs containing 100, 200, and 500 nodes (Table 7), using targeted and hard interventions. For graphs of 100 nodes, the edge number increased by 100 edges per graph for the factor graph dataset, and 1,000 per graph for the spn dataset. In addition to F1 and SHD, we also report the structural intervention distance (SID) Peters & Bühlmann [19] for score-based and Bayesian methods (Table 9 and Table 10).

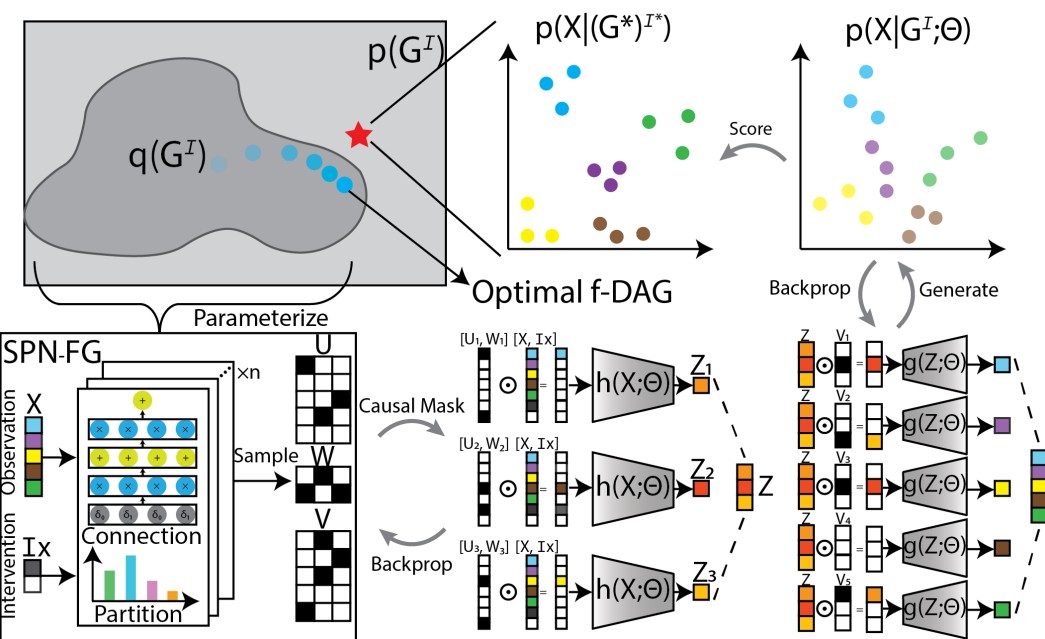

Figure 4: **Overview of ABCDEFG. Top Left:** Bayesian framework. A prior $p$ with a support of all DAGs and a variational distribution with a support of f-DAGs. The red star represents the ground-truth DAG and light blue dots with increasing transparency show an optimization process w.r.t. the variational distribution. **Top Right:** Real vs. generated data distribution. **Bottom:** ABCDEFG model architecture. Binary matrices $\mathbf{U}, \mathbf{V}, \mathbf{W}$ are sampled from a parametric f-DAG model such as SPN-FG. Next, observations are masked by sampled causal relations (under Hadamard product, $\odot$) and fed to a VAE model fitting data distribution. Arrows show direction of data flow and back propagation.

Table 5: Performance on Simulated Datasets with 16 Nodes. Best performance is in bold text and second best is underlined.

| METRIC | METHOD | LINEAR (FG) | | | LINEAR (SPN-FG) | | | NONLINEAR (FG) | | | NONLINEAR (SPN-FG) | | |
|---|---|---|---|---|---|---|---|---|---|---|---|---|---|
| | | D1 | D2 | D3 | D1 | D2 | D3 | D1 | D2 | D3 | D1 | D2 | D3 |
| SHD↓ | DCDI | 12 | 4 | 26 | 14 | 12 | 25 | 14 | **7** | **4** | **2** | 28 | 14 |
| | DCDFG | 48 | 33 | 31 | 43 | 43 | 56 | 48 | 18 | 5 | 46 | 36 | 17 |
| | ENCO | 27 | 24 | 28 | 28 | 54 | 29 | 27 | 18 | 29 | 37 | 41 | 29 |
| | SDCD | 11 | 16 | 3 | 12 | 15 | 5 | 4 | **7** | 6 | 8 | 16 | 5 |
| | ABCDEFG | **0** | **0** | **0** | 12 | **0** | **0** | 2 | 12 | 13 | 3 | 12 | 9 |
| | ABCDEFG (SPN) | **0** | 10 | 12 | **5** | 21 | 1 | 26 | 28 | 26 | 22 | 17 | 25 |
| F1↑ | DCDI | 0.842 | 0.923 | 0.678 | 0.793 | 0.876 | 0.679 | 0.781 | **0.759** | **0.935** | **0.964** | 0.682 | 0.774 |
| | DCDFG | 0.529 | 0.190 | 0.644 | 0.566 | 0.650 | 0.509 | 0.529 | N/A | 0.915 | 0.477 | 0.667 | 0.691 |
| | ENCO | 0.000 | 0.000 | 0.000 | 0.000 | 0.000 | 0.000 | 0.000 | 0.000 | 0.000 | 0.000 | 0.128 | 0.000 |
| | SDCD | 0.825 | 0.750 | 0.949 | 0.818 | 0.842 | 0.918 | 0.931 | **0.759** | 0.889 | 0.833 | 0.795 | **0.915** |
| | ABCDEFG | **1.000** | **1.000** | **1.000** | 0.806 | **1.000** | **1.000** | **0.964** | 0.500 | 0.800 | 0.949 | 0.842 | 0.857 |
| | ABCDEFG (SPN) | **1.000** | 0.828 | 0.824 | **0.912** | 0.753 | 0.983 | 0.675 | 0.333 | 0.690 | 0.718 | 0.805 | 0.683 |

Table 6: F1 score and SHD of Scored methods on Nonlinear Targeted Simulated Datasets with 200 and 500 nodes.

| METRIC | METHOD | HARD INTVN | SOFT INTVN | SPN HARD | SPN SOFT |
|---|---|---|---|---|---|
| F1 (200 NODES) | DCDFG | $0.10 \pm 0.03$ | $0.13 \pm 0.03$ | $0.06 \pm 0.04$ | $0.40 \pm 0.26$ |
| | SDCD | $0.50 \pm 0.08$ | $0.43 \pm 0.06$ | $0.16 \pm 0.01$ | $0.14 \pm 0.01$ |
| | ABCDEFG | $\mathbf{0.52 \pm 0.13}$ | $\mathbf{0.49 \pm 0.07}$ | $0.61 \pm 0.05$ | $0.62 \pm 0.04$ |
| | ABCDEFG (SPN) | $0.48 \pm 0.13$ | $0.42 \pm 0.05$ | $\mathbf{0.62 \pm 0.04}$ | $\mathbf{0.62 \pm 0.03}$ |
| SHD (200 NODES) | DCDFG | $7678 \pm 1846$ | $4954 \pm 1147$ | $13901 \pm 272$ | $10892 \pm 1959$ |
| | SDCD | $\mathbf{517 \pm 38}$ | $592 \pm 16$ | $13634 \pm 368$ | $13854 \pm 516$ |
| | ABCDEFG | $657 \pm 401$ | $583 \pm 285$ | $8978 \pm 966$ | $\mathbf{8739 \pm 743}$ |
| | ABCDEFG (SPN) | $770 \pm 525$ | $\mathbf{583 \pm 169}$ | $\mathbf{8950 \pm 697}$ | $8854 \pm 565$ |
| F1 (500 NODES) | DCDFG | $0.11 \pm 0.10$ | $0.06 \pm 0.00$ | $0.07 \pm 0.04$ | $0.17 \pm 0.10$ |
| | SDCD | $0.34 \pm 0.01$ | $0.32 \pm 0.00$ | $0.10 \pm 0.01$ | $0.09 \pm 0.01$ |
| | ABCDEFG | $\mathbf{0.56 \pm 0.00}$ | $\mathbf{0.55 \pm 0.03}$ | $0.49 \pm 0.05$ | $0.52 \pm 0.06$ |
| | ABCDEFG (SPN) | $0.48 \pm 0.01$ | $0.50 \pm 0.04$ | $\mathbf{0.54 \pm 0.04}$ | $\mathbf{0.56 \pm 0.05}$ |
| SHD (500 NODES) | DCDFG | $22507 \pm 17922$ | $25849 \pm 2023$ | $105723 \pm 2134$ | $99553 \pm 6921$ |
| | SDCD | $1777 \pm 180$ | $1849 \pm 134$ | $105352 \pm 518$ | $105834 \pm 508$ |
| | ABCDEFG | $\mathbf{1262 \pm 25}$ | $\mathbf{1228 \pm 94}$ | $72836 \pm 6061$ | $69401 \pm 7637$ |
| | ABCDEFG (SPN) | $1562 \pm 50$ | $1340 \pm 132$ | $\mathbf{67007 \pm 4940}$ | $\mathbf{64650 \pm 5395}$ |

## C.2 AVAILABILITY OF BENCHMARK RESULTS

We conducted benchmark studies on a variety of data simulation settings at a larger scale, with 100 nodes and 10 factors. We classify the simulations by (1) SEM - linear vs. nonlinear, (2) factor graph model - SPN-FG vs. regular f-DAG and (3) type of intervention (hard vs. soft). We included all results as csv files in our supplementary material. Each csv file records a metric (precision, recall, f1, SHD) for all methods run on one type of simulation. The tables summarized in Table 2 and Table 3 show the mean ± standard deviation for each dataset type, based on the corresponding experimental results. Moreover, the benchmarking results for score-based methods on linear datasets are presented in Fig. 5, as discussed in the main text. In addition, as proof that our model can construct acyclic graphs by design, we calculated the number of cycles when compared with score-based methods (Fig. 6), as well as the number of edges that would need to be removed to obtain an acyclic graph (Fig. 7). Both results suggest that the graphs predicted by our model are naturally acyclic.

Table 7: F1 score and SHD of Scored methods on Nonlinear Targeted Simulated Datasets on dense graphs with hard interventions.

| METRIC | METHOD | NON LINEAR | NON LINEAR SPN |
|---|---|---|---|
| F1 (100 NODES) | DCDI | $0.47 \pm 0.04$ | $0.34 \pm 0.05$ |
| | DCDFG | $0.25 \pm 0.03$ | $0.11 \pm 0.06$ |
| | ENCO | $0.04 \pm 0.00$ | $0.12 \pm 0.06$ |
| | SDCD | $\mathbf{0.66 \pm 0.06}$ | $0.35 \pm 0.06$ |
| | ABCDEFG | $\underline{0.53 \pm 0.07}$ | $\mathbf{0.69 \pm 0.04}$ |
| | ABCDEFG (SPN) | $0.43 \pm 0.05$ | $\underline{0.65 \pm 0.01}$ |
| SHD (100 NODES) | DCDI | $475 \pm 81$ | $3882 \pm 135$ |
| | DCDFG | $1464 \pm 910$ | $3866 \pm 98$ |
| | ENCO | $2185 \pm 223$ | $3982 \pm 228$ |
| | SDCD | $\mathbf{245 \pm 30}$ | $3283 \pm 148$ |
| | ABCDEFG | $\underline{567 \pm 92}$ | $\mathbf{1909 \pm 182}$ |
| | ABCDEFG (SPN) | $770 \pm 525$ | $\underline{2132 \pm 59}$ |
| F1 (200 NODES) | DCDFG | $0.18 \pm 0.06$ | $0.11 \pm 0.02$ |
| | SDCD | $0.42 \pm 0.05$ | $0.18 \pm 0.02$ |
| | ABCDEFG | $\mathbf{0.56 \pm 0.08}$ | $\underline{0.59 \pm 0.03}$ |
| | ABCDEFG (SPN) | $\underline{0.49 \pm 0.06}$ | $\mathbf{0.60 \pm 0.01}$ |
| SHD (200 NODES) | DCDFG | $6933 \pm 2978$ | $17035 \pm 257$ |
| | SDCD | $\mathbf{885 \pm 168}$ | $16575 \pm 142$ |
| | ABCDEFG | $\underline{932 \pm 408}$ | $\underline{9817 \pm 600}$ |
| | ABCDEFG (SPN) | $1130 \pm 445$ | $\mathbf{9649 \pm 104}$ |
| F1 (500 NODES) | DCDFG | $0.09 \pm 0.07$ | $0.03 \pm 0.03$ |
| | SDCD | $\underline{0.26 \pm 0.01}$ | $0.10 \pm 0.00$ |
| | ABCDEFG | $\mathbf{0.33 \pm 0.05}$ | $\underline{0.45 \pm 0.04}$ |
| | ABCDEFG (SPN) | $0.25 \pm 0.06$ | $\mathbf{0.47 \pm 0.03}$ |
| SHD (500 NODES) | DCDFG | $\underline{4298 \pm 131}$ | $118625 \pm 1433$ |
| | SDCD | $\mathbf{4294 \pm 191}$ | $114183 \pm 421$ |
| | ABCDEFG | $6598 \pm 1804$ | $\underline{80315 \pm 5300}$ |
| | ABCDEFG (SPN) | $8662 \pm 3101$ | $\mathbf{77089 \pm 3660}$ |

## C.3 EXPERIMENT SETTINGS

In this section, we report the hyperparameters used in our simulation study. Because ABCDEFG has many hyperparameters, we did not comprehensively tune each of them. Instead, we fixed hyperparameters across the same SEM model type. Here, we report some key hyperparameter values. For the other hyperparameters, our python program contains default values and we used the same value in all experiments. Table 11 summarizes the most important hyperparameters. In addition, we unexhaustively tuned the L1 regularization coefficient by trying two different values per simulation type. We also have a separate L1 regularization coefficient for the intervention-to-node bipartite graph in simulation with unknown intervention targets.

Table C.3 lists the set of best parameters we chose for each simulation type. For conciseness, we name a simulation type by a sequence of four attributes: (1) targeted (T) vs. untargeted (U), (2) canonical f-DAG (FG) vs. SPN-FG (SPNFG), (3) linear (L) vs nonlinear (N) SEM, and (4) hard (H) vs. soft (S) intervention, separated by "-".

Table 8: Precision and recall of Bayesian methods on Simulated Datasets with 16 Nodes.

| METRIC | METHOD | LINEAR FG | LINEAR SPNFG | NONLINEAR FG | NONLINEAR SPNFG |
|---|---|---|---|---|---|
| PRECISION | BaCaDi | $0.11 \pm 0.01$ | $0.15 \pm 0.03$ | $0.10 \pm 0.02$ | $0.13 \pm 0.02$ |
| | DECI | $0.11 \pm 0.04$ | $0.17 \pm 0.01$ | $0.09 \pm 0.03$ | $0.12 \pm 0.04$ |
| | VI-DP-DAG | $0.14 \pm 0.02$ | $0.14 \pm 0.03$ | $0.08 \pm 0.01$ | $0.15 \pm 0.05$ |
| | PRODAG | $0.11 \pm 0.01$ | $0.13 \pm 0.02$ | $0.10 \pm 0.01$ | $0.15 \pm 0.04$ |
| | ABCDEFG | $\mathbf{0.77 \pm 0.05}$ | $\mathbf{0.51 \pm 0.13}$ | $\mathbf{0.31 \pm 0.25}$ | $\mathbf{0.54 \pm 0.12}$ |
| | ABCDEFG (SPN) | $0.37 \pm 0.04$ | $0.28 \pm 0.06$ | $0.17 \pm 0.10$ | $0.35 \pm 0.19$ |
| RECALL | BaCaDi | $0.48 \pm 0.02$ | $\mathbf{0.51 \pm 0.02}$ | $\mathbf{0.44 \pm 0.01}$ | $\mathbf{0.46 \pm 0.01}$ |
| | DECI | $0.07 \pm 0.02$ | $0.09 \pm 0.01$ | $0.06 \pm 0.02$ | $0.06 \pm 0.02$ |
| | VI-DP-DAG | $0.47 \pm 0.05$ | $0.36 \pm 0.04$ | $0.30 \pm 0.03$ | $0.38 \pm 0.07$ |
| | PRODAG | $0.44 \pm 0.01$ | $0.43 \pm 0.02$ | $0.35 \pm 0.02$ | $0.47 \pm 0.06$ |
| | ABCDEFG | $\mathbf{0.74 \pm 0.20}$ | $0.48 \pm 0.13$ | $0.23 \pm 0.32$ | $0.30 \pm 0.23$ |
| | ABCDEFG (SPN) | $0.44 \pm 0.03$ | $0.24 \pm 0.08$ | $0.12 \pm 0.15$ | $0.29 \pm 0.26$ |

Table 9: SID of Bayesian methods on Non linear Simulated Datasets with 16 Nodes.

| METRIC | METHOD | NON LINEAR | NON LINEAR SPN |
|---|---|---|---|
| SID | DECI | $65.76 \pm 34.86$ | $109.32 \pm 18.75$ |
| | VI-DP-DAG | $83.52 \pm 35.83$ | $92.19 \pm 7.59$ |
| | PRODAG | $62.01 \pm 25.20$ | $90.1 \pm 21.78$ |
| | ABCDEFG | $\mathbf{41.93 \pm 27.27}$ | $\mathbf{64.33 \pm 15.75}$ |
| | ABCDEFG (SPN) | $50.72 \pm 27.49$ | $70.85 \pm 24.81$ |

Table 10: SID of score-based methods on Non linear Simulated Datasets with 100 Nodes.

| METRIC | METHOD | HARD INTVN | SOFT INTVN | SPN HARD | SPN SOFT |
|---|---|---|---|---|---|
| SID | DCDFG | $1839 \pm 308$ | $1595 \pm 1418$ | $5860 \pm 1662$ | $6976 \pm 346$ |
| | ENCO | $3668 \pm 864$ | $3722 \pm 945$ | $8805 \pm 221$ | $8899 \pm 200$ |
| | SDCD | $2189 \pm 616$ | $2224 \pm 1009$ | $6843 \pm 504$ | $6858 \pm 381$ |
| | ABCDEFG | $1005 \pm 327$ | $771 \pm 438$ | $\mathbf{4615 \pm 681}$ | $\mathbf{4710 \pm 694}$ |
| | ABCDEFG (SPN) | $\mathbf{809 \pm 510}$ | $\mathbf{671 \pm 414}$ | $4724 \pm 595$ | $4783 \pm 594$ |

## C.4 TIME AND MEMORY CONSUMPTION

All simulated datasets with known intervention targets contain 25k samples and those with unknown intervention targets contain 30k samples. With a batch size of 128, we were able to train our model on a server with 2 2x 2.9 GHz Intel Xeon Gold 6226R, 16 GB of RAM and an NVIDIA A40 GPU with 48GB of memory. The training time of ABCDEFG is shown in Fig. 8. Since datasets are of similar sizes, the training time is stable across different simulations. Training ABCDEFG with SPN-FG consumes more time due to a larger number of parameters and extra time for forward and backward through the network layers. The benchmarking of Bayesian methods was conducted on datasets with 16 nodes. The training times for the different methods are shown in Table 14. All methods, except BaCaDi, were run on an NVIDIA A40 GPU with 16GB of RAM. (No GPU implementation was available for BaCaDi.)

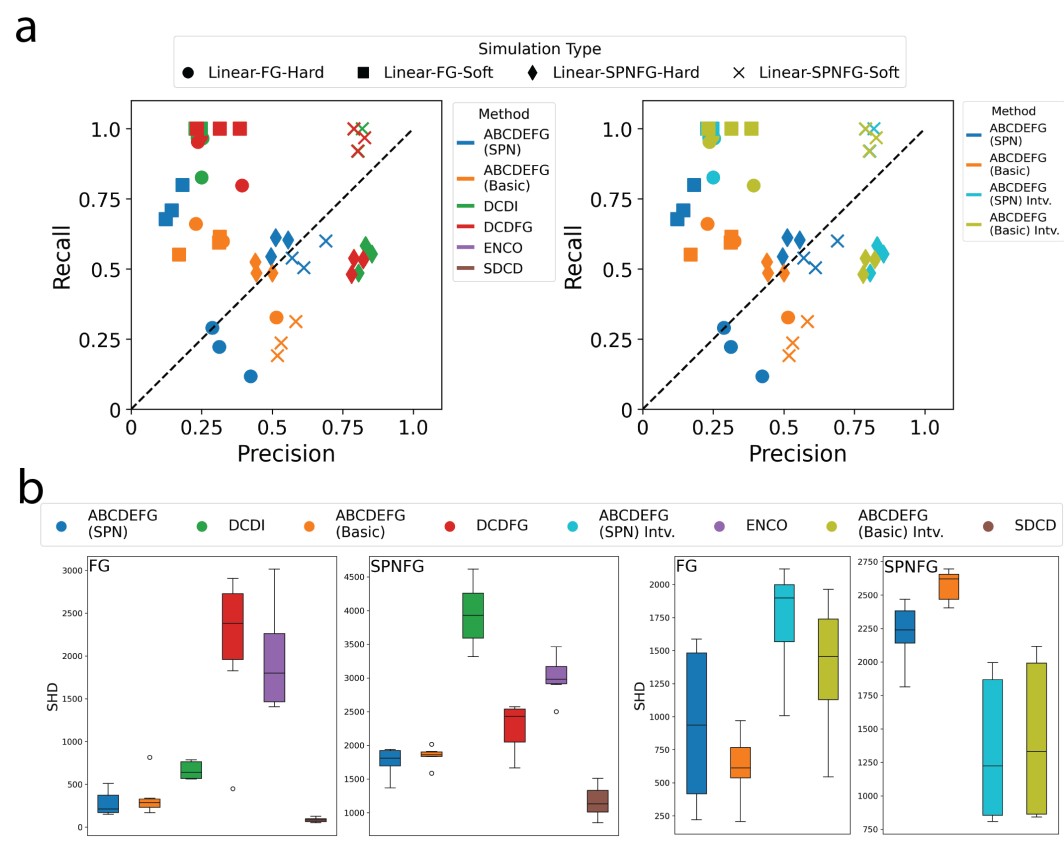

Figure 5: **Benchmarking of score-based methods on linear datasets.** (a) Precision and recall for different score based methods, dataset types are shown in different shapes.(b) SHD comparison between different score based methods on targeted datasets(left two), and untargeted datasets (right two).

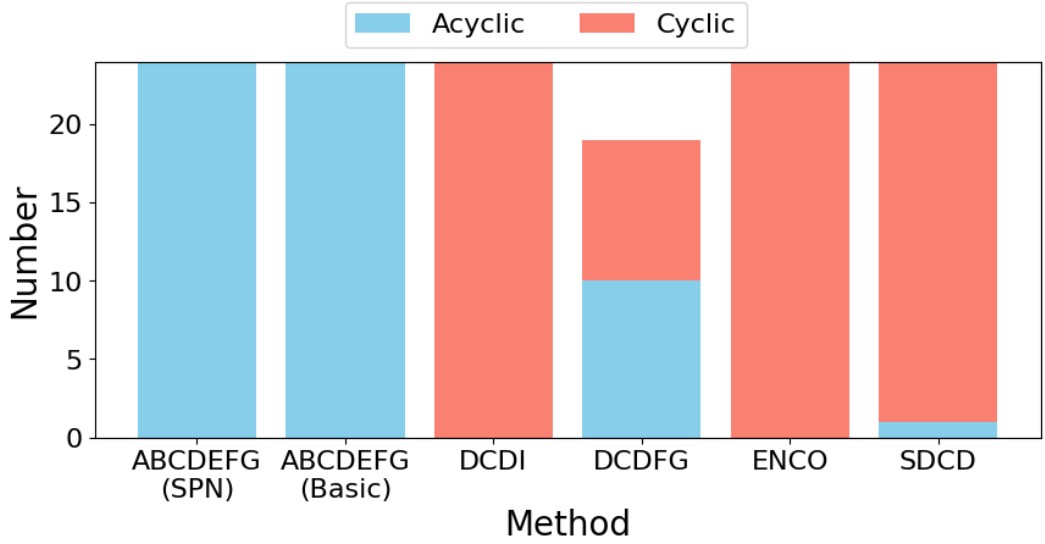

Figure 6: **Comparison of number of acyclic and cyclic graphs.**

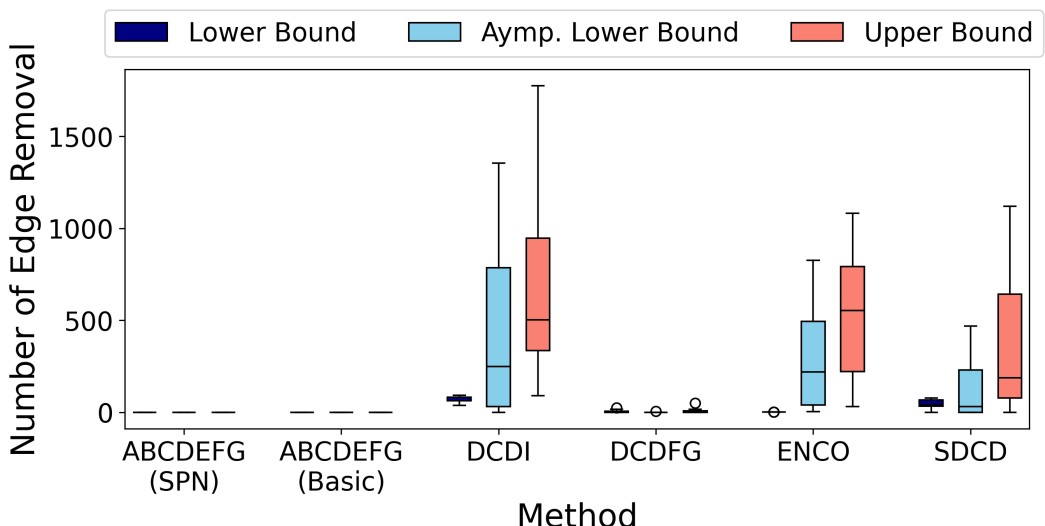

Figure 7: **Comparison of to be removed number of edges for acyclic graphs.** Upper and lower bound of number of to be removed edges are colored in blue and red, respectively.

Table 11: Default Hyper-Parameter Setting of ABCDEFG in a Simulation Study.

| PARAMETER NAME | DEFAULT VALUE |
|---|---|
| BATCH SIZE | 128 |
| HIDDEN DIMENSION | 1000 |
| NUMBER OF EPOCHS | 1000 |
| NUMBER OF HIDDEN LAYERS | 1 |
| WIDTH BOUND OF SPN (MAX_COPIES) | 8 |
| LEARNING RATE (VAE) | $5 \times 10^{-4}$ |
| LEARNING RATE (F-DAG MODEL) | $5 \times 10^{-3}$ |
| KL DIV. COEFF. ($\beta$) | $1 \times 10^{-8}$ |
| GAUSSIAN NOISE LEVEL | 0.05 |
| VAE WEIGHT L2 REG. | $1 \times 10^{-3}$ |
| LATENT FACTOR PRIOR | $\mathcal{N}(\mathbf{0}, 10^{-3} \cdot \mathbf{I})$ |

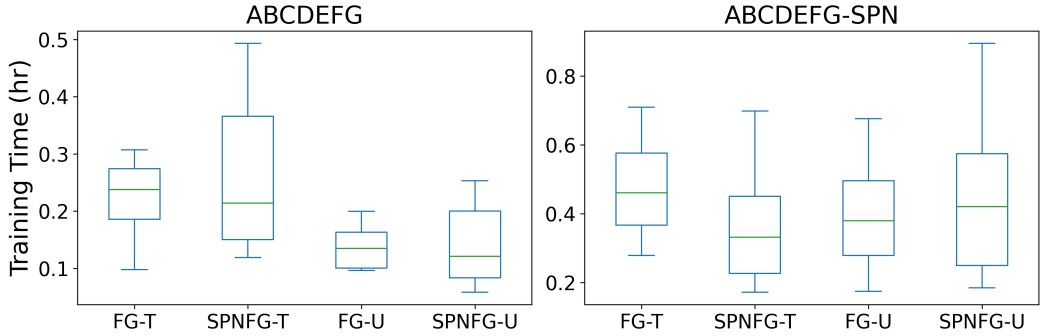

Figure 8: **Training Time of ABCDEFG.** Each box represents one type of simulation. We groups simulation regarding the ground truth graph type and known vs. unknown intervention targets. We use the suffix "-T" for known intervention targets and "-U" for unknown ones.

Table 12: Hyper-Parameter Setting of ABCDEFG in a Simulation Study.

| SIMULATION TYPE | L1 REG. | L1 REG. (INTV.) | ACTIVATION FUNCTION | SPN PARALLELISM |
|---|---|---|---|---|
| T-FG-L-H | 0.1, 0.1 | N/A | IDENTITY | NODE |
| T-FG-L-S | 0.01, 0.01 | N/A | IDENTITY | FACTOR |
| T-FG-N-H | 1.0, 1.0 | N/A | TANH | FACTOR |
| T-FG-N-S | 0.01, 0.001 | N/A | TANH | NODE |
| T-SPNFG-L-H | 0.01, 0.01 | N/A | IDENTITY | NODE |
| T-SPNFG-L-S | 1E-4, 1E-4 | N/A | IDENTITY | NODE |
| T-SPNFG-N-H | 0.01, 0.01 | N/A | TANH | NODE |
| T-SPNFG-N-S | 1E-4, 1E-4 | N/A | TANH | FACTOR |
| U-FG-L-H | 0.01, 0.01 | 10.0, 10.0 | IDENTITY | NODE |
| U-FG-L-S | 1E-4, 1E-4 | 10.0, 10.0 | IDENTITY | NODE |
| U-FG-N-H | 0.01, 0.01 | 10.0, 10.0 | TANH | NODE |
| U-FG-N-S | 1E-4, 1E-4 | 10.0, 10.0 | TANH | NODE |
| U-SPNFG-L-H | 1E-6, 1E-6 | 0.1, 0.1 | IDENTITY | FACTOR |
| U-SPNFG-L-S | 1E-7, 1E-7 | 1.0, 1.0 | IDENTITY | NODE |
| U-SPNFG-N-H | 1E-6, 1E-6 | 0.1, 0.1 | TANH | FACTOR |
| U-SPNFG-N-S | 1E-8, 1E-7 | 1.0, 1.0 | TANH | NODE |

Table 13: Literature Overview

| METHOD | UNKNOWN TARGET IDENTIFICATION | LIKELIHOOD COMPLEXITY | DAG PENALTY COMPLEXITY | SPACE COMPLEXITY | DAG GUARANTEE BY CONSTRUCTION |
|---|---|---|---|---|---|
| DCDI | PARTIAL | $O(n^2)$ | $O(n^3)$ | $O(n^2)$ | NO |
| DCDFG | NO | $O(mn)$ | $O(mn)$ | $O(mn)$ | NO |
| ENCO | NO | $O(n^2)$ | N/A | $O(n^2)$ | NO |
| SDCD | NO | $O(n^2)$ | $O(n^2)$ | $O(n^2)$ | NO |
| ABCDEFG | YES | $O(mn)$ | N/A | $O(mn)$ | YES |

Table 14: Time usage on Simulated Datasets with 16 Nodes.

| METHOD | LINEAR FG | LINEAR SPNFG | NONLINEAR FG | NONLINEAR SPNFG |
|---|---|---|---|---|
| BACADI | $1704.56 \pm 33.70$ | $1405.64 \pm 277.53$ | $1265.71 \pm 35.65$ | $1435.08 \pm 20.63$ |
| DECI | $987.81 \pm 5.72$ | $985.27 \pm 1.64$ | $994.50 \pm 0.91$ | $991.40 \pm 0.30$ |
| VI-DP-DAG | $764.77 \pm 255.60$ | $245.63 \pm 132.17$ | $501.51 \pm 328.49$ | $302.64 \pm 180.52$ |
| PRODAG | $79.37 \pm 0.76$ | $79.99 \pm 2.34$ | N/A | N/A |
| ABCDEFG | $82.60 \pm 24.36$ | $65.95 \pm 24.61$ | $70.51 \pm 34.89$ | $106.71 \pm 68.05$ |
| ABCDEFG (SPN) | $138.64 \pm 37.63$ | $67.63 \pm 32.89$ | $136.17 \pm 52.82$ | $177.71 \pm 154.83$ |

## D PREPROCESSING SINGLE CELL PERTURBATION DATA

The data used for single cell perturbation is downloaded from Amin et al. [2] and we followed the preprocessing steps described by Lopez et al. [17]. For each untargeted perturbation, we removed the description words like 'high','low','early',eta, and only retain the name of each biomolecule as the perturbation. We used scanpy to select the top 1000 highly variable genes as input of our model, and used 10 factors. We performed gene ontology analysis using the online tool at the Gene Ontology Website.

# E   LARGE LANGUAGE MODELS (LLM) USAGE STATEMENT

We use LLM as a tool for assisting paper writing and sentences polishing, and not for generating or contributing ideas related to this paper.

