# OpenReview forum: "Amortized Bayesian Causal Discovery of Extended Factor Graphs"
_ICLR.cc/2026/Conference — Submitted to ICLR 2026_

### Official Review · Reviewer_7yqf · 2025-10-28

**Soundness:** 3
**Presentation:** 3
**Contribution:** 2
**Rating:** 6
**Confidence:** 2

**Summary:**

This paper presents a new method, called ABCDEFG, for learning causal relationships from experimental (interventional) data with unkown targets. The key idea is to introduce the extended factor graph, which can handle uncertainty in which interventions act on which variables. The paper claims strong theoretical guarantees (identifiability) and demonstrates superior performance on both simulated datasets and a real-world biological dataset.

**Strengths:**

1. The proposed method is scalable, identifiable, and can handle uncertainty estimation.

2. The experiments show consistent improvements in F1 score and SHD over many other methods on the synthetic dataset and lower MSE in the real-world dataset.

3. The problem addressed is important, as learning causal relationships from complex biological data remains a significant and challenging task.

**Weaknesses:**

So far, the largest question from my side is the lack of motivation for the proposed framework. I’m not very familiar with the factor graph framework, so I’d appreciate more explanation on the motivation for using it in this paper. In causal discovery, most existing methods are based on DAGs or I-DAGs (for interventional data), while this paper follows the approach of Lopez by extending factor graphs to handle biological data. I’m curious to understand what unique advantages factor graphs offer in this setting, especially given that biological data often involve selection bias and latent variables. Is the idea that factors can, to some extent, represent latent variables or capture dependencies that DAGs cannot?

Another question is about the identifiability. I appreciate that the authors included a proof sketch, which nicely conveys the general intuition. However, it’s still difficult for me to grasp the role of each assumption. Specifically, how does each assumption contribute to establishing the identifiability result?

**Questions:**

Could the authors briefly discuss the interpretability of the learned causal factor graph shown in Figure 2(b)? It would be helpful to understand whether the discovered structure offers any biologically meaningful insights or aligns with known domain knowledge.

---

> ### Author Response · Authors · 2025-11-28
>
> # Motivation for using factor DAGs
> A key intuition behind our approach is that interventions performed on biological cells tend to change entire groups of genes (biological pathways or processes). The factor graph assumption directly captures this intuition. From this starting point, we want to build an approach that can simultaneously identify which genes regulate other genes (gene-to-gene causal relationships) and which genes are altered by the interventions (intervention-to-gene causal relationships). Both types of causal relationships are important for learning from the latest biological experiments, in which the results of many interventions can be measured with single-cell RNA sequencing. Incorporating interventions to learn gene-gene relationships is also very important for biological applications, because it is well established that observational measurements alone are insufficient for learning gene regulatory mechanisms.
>
> Furthermore, the f-DAG assumption is natural because gene regulatory networks have hierarchical organization and are often represented in terms of low-rank summaries such as pathways, factors, etc. The DCDFG paper (Lopez et al.) presented a theoretical argument justifying the factor graph framework. Briefly, an n-by-n adjacency matrix can be factorized as a Boolean product of $n$-by-$m$ and $m$-by-$n$ matrices where $m < n$. At the graph level, this means edge connections can be bundled together into m factors. Lopez et al. proved that the Boolean rank of such a matrix increases with high probability if an edge is randomly perturbed. This motivates the use of a factor graph as a sort of low-rank constraint – incorrect edges result in a higher Boolean rank with high probability. Conversely, learning a factor graph makes the causal inference process more robust to noisy edges–a big problem in noisy biological data. An additional benefit is that the factors are biologically interpretable in terms of pathways, and can even be compared to curated pathway annotations such as the Gene Ontology.
> Consistent with this theoretical justification, our experiments indicate that this factor graph approach provides a significant performance boost, particularly when there is higher-order structure in the network (the SPN simulations mentioned above).
>
> The factor graph assumption also provides us with a tractable theoretical framework for building a causal inference approach with desirable properties. There are multiple benefits to using an f-DAG. (1) The factor DAG provides a way to learn causal graphs that are acyclic by construction. (2) The f-DAG framework provides a way to effectively and efficiently learn intervention-to-gene and gene-to-gene relationships from interventions with unknown targets. (3) The factor DAG framework provides a way to reason about the identifiability of the causal DAG.
>
> # Role of assumptions in identifiability proof
> he four assumptions were originally used in the DCDI paper to establish an identifiability guarantee for differentiable causal discovery with interventions. Our proof here follows a similar argument to theirs.
>
> 1. Sufficient capacity makes sure our model parameter space is large enough to include the true distributions. The use of deep neural networks should fulfill this assumption.
>
> 2. I-faithfulness means (1) all the conditional independence is due to graph structure (d-separation) and (2) there is no pathological intervention, so that any intervened node should have a different distribution. This assumption is key to make sure optimizing the joint distribution is consistent with inferring the causal graph.
>
> 3. Positivity means the family of joint distributions we consider has a non-zero pdf/pmf everywhere conditioned on its parents. This property is used in the proof of a lemma for identifiability guarantee. The lemma says if two non-Markov-equivalent causal graphs generate two joint distributions p and f respectively and p is positive everywhere, then KL(p, f) > 0 .
>
> 4. Finite differential entropy means the joint distribution under a causal graph and intervention should not be heavy-tailed. This assumption is used to prove another lemma.
>
> These four assumptions are necessary for proving the score (likelihood of all observations given a fixed causal graph) of a graph in the true I-Markov equivalence class (I-MEC) is always higher than the score of any other graph not in the true I-MEC. Our identifiability theorem considers a Bayesian setting and we prove the ELBO of any graph in the true I-MEC is greater than that of any other graph not in the true I-MEC. We use all four assumptions because the ELBO has one term containing the score from the DCDI and we use DCDI’s theorem to prove ours.

---

> > ### Author Response · Authors · 2025-11-28
> >
> > # Interpretability of factor graph in Fig. 2b
> > Thanks for giving us a chance to clarify. Figure 2b does indeed highlight the interpretability of the factor graph we learn.
> > The pentagons represent perturbations (intervention with unknown targets), and the circles represent causal graph factors. The following types of directed edges are learned from our model: intervention to latent factor, gene to latent factor, and latent factor to gene. From this panel, we want to show the following points:
> >
> > (1) Our model learns perturbation-specific responses through the representation by factors. For example, our model learned the edges from bmp4 and bmp7 perturbations towards factor 2, and through our gene ontology analysis, factor 2 is enriched in terms like response to BMP, which suggests this factor correctly identifies genes in the BMP signaling pathway.
> >
> > (2) Our model learns the regulation relationships between upstream and downstream genes. For example, we colored the
> > text of some genes that have edges from genes to factors in blue in this panel, and we can see most of the genes are transcription factors. This makes sense biologically, because transcription factors are the most common type of upstream regulators in gene regulatory networks. Similarly, we colored the text of genes which have edges from factors to genes in orange. Many of these genes coding proteins like structural proteins and enzymes which usually serve as the terminal effectors of a biological process.
> >
> > We also included this figure simply because we think that it helps to communicate the type of causal graph (extended factor graph) learned by our approach and its interpretation in terms of biological mechanisms.

---

### Official Review · Reviewer_e9Uu · 2025-10-30

**Soundness:** 2
**Presentation:** 1
**Contribution:** 2
**Rating:** 2
**Confidence:** 3

**Summary:**

This paper argues for the lack of causal discovery methods that can handle thousands of variables and can incorporate intervention with unknown targets.  To address these limitations, they developed the Amortized Bayesian Causal Discovery of Extended Factor Graphs (ABCDEFG) to approximate the causal graph by the f-DAG under the differentiable Bayesian framework. Experimental results on simulated and real-world single-cell perturbation data verify its effectiveness.

**Strengths:**

1. This paper proposed ABCDEFG, which can do causal discovery for large-scale data with unknown intervention targets.
2. They approximate causal discovery with f-DAG under the differentiable Bayesian framework.

**Weaknesses:**

1. It takes me a long time to understand the contribution and writing logic. The motivation in the first two paragraphs is the need for high-dimensional causal discovery. However, all the contribution is related to modeling intervention. Intervention provides extra information, but how does it help causal discovery in factor-DAG?
2. The contribution is the Amortized Bayesian Causal Discovery in Section 2.5. However, I did not see the direct benefit of assuming the causal graph can be approximated by the f-DAG under the differentiable Bayesian framework.
3. The model of the ABCDEFG is in the appendix.

**Questions:**

1. What does the symbol $\Lambda$ mean in Line 241.
2. Line 252: in the Appendix is not clear.
3. Equation (1) is the same as the one below it (Line 269)
4. Only Equation (1) has the equation number, while others do not.
5. The setting for gene expression is not strong enough. We know that the gene regulatory network is a complex system with latent cofounders like non-coding RNAs and biological constraints. However, the ABVDEFG model has strong assumptions, including DAG, causal sufficiency, and no selection bias. Some work can handle latent confounders with unknown intervention targets, like Causal Discovery from Soft Interventions with Unknown Targets: Characterization and Learning.
6. Score-based causal discovery can handle thousands of genes as well, like FGES, A million variables and more: the Fast Greedy Equivalence Search algorithm for learning high-dimensional graphical causal models, with an application to functional magnetic resonance images.

---

> ### Author Response · Authors · 2025-11-28
>
> # Motivation and rationale for incorporating interventions into factor DAG approach
> A key intuition behind our approach is that interventions performed on biological cells tend to change entire groups of genes (biological pathways or processes). The factor graph assumption directly captures this intuition. From this starting point, we want to build an approach that can simultaneously identify which genes regulate other genes (gene-to-gene causal relationships) and which genes are altered by the interventions (intervention-to-gene causal relationships). Both types of causal relationships are important for learning from the latest biological experiments, in which the results of many interventions can be measured with single-cell RNA sequencing. Incorporating interventions to learn gene-gene relationships is also very important for biological applications, because it is well established that observational measurements alone are insufficient for learning gene regulatory mechanisms.
> The factor graph assumption also provides us with a tractable theoretical framework for building a causal inference approach with desirable properties (including guaranteed acyclicity, distributional estimation, and identifiability guarantees).
>
> # Benefit of factor DAG assumption
> Gene regulatory networks have hierarchical organization and are often represented in terms of low-rank summaries such as pathways, factors, etc. The DCDFG paper (Lopez et al.) presented a theoretical argument justifying the factor graph framework. Briefly, an n-by-n adjacency matrix can be factorized as a Boolean product of $n$-by-$m$ and $m$-by-$n$ matrices where $m < n$. At the graph level, this means edge connections can be bundled together into m factors. Lopez et al. proved that the Boolean rank of such a matrix increases with high probability if an edge is randomly perturbed. This motivates the use of a factor graph as a sort of low-rank constraint – incorrect edges result in a higher Boolean rank with high probability. Conversely, learning a factor graph makes the causal inference process more robust to noisy edges–a big problem in noisy biological data. An additional benefit is that the factors are biologically interpretable in terms of pathways, and can even be compared to curated pathway annotations such as the Gene Ontology.
> Consistent with this theoretical justification, our experiments indicate that this factor graph approach provides a significant performance boost, particularly when there is higher-order structure in the network (the SPN simulations mentioned above).
> Practically speaking, using a f-DAG provides the theoretical machinery we need to (1) learn causal graphs that are acyclic by construction and (2) effectively and efficiently learn intervention-to-gene and gene-to-gene relationships from interventions with unknown targets.
>
> # The model of the ABCDEFG is in the appendix.
> Due to the page limit, we had to put some details about the ABCDEFG model in the appendix, but we describe the core part and the most important results in the main text.
>
> # Lambda in Line 241.
> The symbol Lambda refers to the set of all parameters we use for a variational graph distribution, q.
>
> # Line 252 in the Appendix
> We provided a derivation of ELBO in Appendix section B.2. There is a slight difference where we added a graph regularization term in appendix B.2.
>
> # Line 269
> Thanks for pointing this out. We will remove line 269 in the text.
>
> # Missing equation numbers
> Thanks for pointing this out. We will reindex the equation numbers.
>
> # Model assumptions
> The DAG and causal sufficiency assumptions we use are not unique to our paper, but are standard in the field. The other state-of-the-art models that we compare against also make these assumptions. The paper that the reviewer cites does not show any results on real data, so it’s hard to assess how practical it is in a real-world setting. From our survey of the literature, there are still no methods with the combination of properties ABCDEFG has, including incorporating interventions with unknown targets, guaranteed acyclicity, Bayesian estimation, and scalability. Extending our framework to incorporate latent confounders is an exciting direction for future work, but one has to start somewhere. ABCDEFG is an important step forward and allows new types of analyses in real-world computational biology applications. We don’t think one needs to solve all problems in order to make a useful contribution.
>
> #Scalability of score-based causal discovery methods
> We agree with the reviewer–the paper cited does show that score-based causal discovery can be scaled up. We will rephrase our introduction and cite this paper.

---

### Official Review · Reviewer_97Bh · 2025-11-01

**Soundness:** 2
**Presentation:** 2
**Contribution:** 2
**Rating:** 4
**Confidence:** 4

**Summary:**

The paper provides a new approach to causal discovery that's scalable, able to quantify uncertainty, capable of utilizing interventional data over unknown targets, and comes with identifiability guarantees.

**Strengths:**

- generally clearly written
- more general, efficient, and performant than existing deep generative models approaches to DAG learning
- theoretically nice approach to DAG learning, as far as deep generative models go: the paper cleanly presents the factor graph set-up from the literature, extends it to the general/unknown interventional setting, and derives the ELBO.

**Weaknesses:**

- not clear how practical the method actually is: the motivating gene regulatory network application doesn't really match the low-rank, causally sufficient DAG set-up (or if so, can the authors justify this?); also not clear how badly performance degrades when these assumptions are violated.
- unnormalized SHD in the tables are hard to interpret, especially without clear possible ranges and 'chance' baselines for comparison---consider [1]; F1 also hard to interpret without some 'chance' baseline when classes are unbalanced (i.e., edge probability not close to 0.5).
- no mention of or comparison against more traditional (i.e., MCMC-based) Bayesian approaches for causal discovery; there's a lot to choose from, going back to [2], but perhaps something more recent like a method in [3] is a reasonable starting point.
- unpolished writing, e.g., Figure 2 is illegible, math formatting throughout Section 2 is off ($KL$ instead of $\mathrm{KL}$, comma instead of colon in definition of $\Xi$, using $p(a|b)$ instead of $p(a\mid b)$, likewise for $\|$), after defining eq. (1) on L247, it appears to be identically repeated only 20 lines later, references [11] and [12] in the paper are duplicates

[1] Petersen, A. H. (2025) Are You Doing Better Than Random Guessing? A Call for Using Negative Controls When Evaluating Causal Discovery Algorithms. In The 41st Conference on Uncertainty in Artificial Intelligence.

[2] Heckerman, D., Meek, C., & Cooper, G. (1997). A Bayesian approach to causal discovery. Technical report msr-tr-97-05, Microsoft Research.

[3] Suter, P., Kuipers, J., Moffa, G., & Beerenwinkel, N. (2023). Bayesian Structure Learning and Sampling of Bayesian Networks with the R Package BiDAG. Journal of Statistical Software, 105(9), 1–31. https://doi.org/10.18637/jss.v105.i09

**Questions:**

I'm happy to increase my rating, depending on the answers to these questions:
1. Can the authors address the weakness mentioned above about real-world applicability and robustness to assumption violations?
2. Likewise, how about the metrics reported in the tables?
2. Likewise, how about discussing and comparing to some MCMC methods?
3. The definition on L147 seems to have removed the interventional Markov factorization described above on L137. Is this intentional?
4. The unspecified "equivalence class" mentioned on pages 1 and 2 is the interventional Markov equivalence class described later, right? Better to use the precise phrase upfront (and fine to leave the definition for later).
5. Is the $m$ in Table 1 the number of factor nodes? These aren't described until page 4, and then they're still not explicitly connected to the usage in Table 1.

---

> ### Author Response · Authors · 2025-11-28
>
> We thank the reviewer for the thoughtful comments and would like to clarify several points in response.
>
> # Rationale for factor graph assumption
> Actually, the gene regulatory network setup nicely matches the low-rank, causally sufficient DAG setup. Gene regulatory networks have hierarchical organization and are often represented in terms of low-rank summaries such as pathways, factors, etc. The DCDFG paper (Lopez et al.) presented a theoretical argument justifying the factor graph framework. Briefly, an n-by-n adjacency matrix can be factorized as a Boolean product of $n$-by-$m$ and $m$-by-$n$ matrices where $m < n$. At the graph level, this means edge connections can be bundled together into m factors. Lopez et al. proved that the Boolean rank of such a matrix increases with high probability if an edge is randomly perturbed. This motivates the use of a factor graph as a sort of low-rank constraint – incorrect edges result in a higher Boolean rank with high probability. Conversely, learning a factor graph makes the causal inference process more robust to noisy edges–a big problem in noisy biological data. An additional benefit is that the factors are biologically interpretable in terms of pathways, and can even be compared to curated pathway annotations such as the Gene Ontology.
> Consistent with this theoretical justification, our experiments indicate that this factor graph approach provides a significant performance boost, particularly when there is higher-order structure in the network (the SPN simulations mentioned above).
>
> # Random baseline for F1 and SHD
> We added a chance baseline as suggested in reference [1]. To control for variation in edge density, we ensure that the randomly generated DAGs have the same number of edges as the true DAG. These results indicate that, in all cases, we significantly outperform the chance baseline in terms of F1 and SHD.
> Here is the revised Table 2:
> Table 2: F1 score and SHD of Scored methods on Nonlinear Targeted Simulated Datasets
> F1 score
> | METHOD | HARD INTVN | SOFT INTVN |SPN HARD | SPN SOFT |
> | :----- | :-----: | :-----: |:-----: | :-----: |
> |RANDOM| 0.04 ± 0.01 |0.03 ± 0.01 | 0.29 ± 0.04 |0.28 ± 0.02 |
> |DCDI| 0.19 ± 0.05 |0.25 ± 0.07 | 0.34 ± 0.01 |0.35 ± 0.04 |
> |DCDFG| 0.05 ± 0.08 |0.20 ± 0.14 | 0.23 ± 0.18 |0.57 ± 0.14 |
> |ENCO| 0.10 ± 0.01 |0.10 ± 0.03 | 0.25 ± 0.01 |0.23 ± 0.03 |
> |SDCD| 0.31 ± 0.01 |0.30 ± 0.06 | 0.25 ± 0.02 |0.30 ± 0.06 |
> |ABCDEFG| 0.29 ± 0.03 |0.25 ± 0.01 | 0.64 ± 0.01 |0.61 ± 0.03 |
> |ABCDEFG (SPN)| 0.29 ± 0.04 |0.21 ± 0.01 | 0.61 ± 0.02 |0.60 ± 0.02 |
>
> SHD
> | METHOD | HARD INTVN | SOFT INTVN |SPN HARD | SPN SOFT |
> | :----- | :-----: | :-----: |:-----: | :-----: |
> |RANDOM| 607 ± 115 |611 ± 111 | 3198 ± 99 |3236 ± 64 |
> |DCDI| 740 ± 291 |599 ± 106 | 4293 ± 301 | 3337 ± 120 |
> |DCDFG| 2513 ± 0 |900 ± 272 | 2500 ± 198 |2030 ± 125 |
> |ENCO| 1952 ± 126 | 1992 ± 141 | 2855 ± 177 | 2896 ± 100 |
> |SDCD| 421 ± 77 | 421 ± 78 | 2973 ± 72 | 2793 ± 83 |
> |ABCDEFG| 1114 ± 328 |1406 ± 361 | 2046 ± 49 | 2248 ± 200 |
> |ABCDEFG (SPN)| 1125 ± 248 | 1791 ± 249 | 2206 ± 81 |2228 ± 85 |
>
> Figure 3 (Nonlinear Simulated Datasets with Unknown Targets)
>
> F1 score
> | METHOD | HARD INTVN | SOFT INTVN |SPN HARD | SPN SOFT |
> | :----- | :-----: | :-----: |:-----: | :-----: |
> |RANDOM| 0.03 ± 0.01 |0.03 ± 0.00 | 0.28 ± 0.02 |0.26 ± 0.03 |
> |ABCDEFG| 0.23 ± 0.01 |0.23 ± 0.05| 0.22 ± 0.06 |0.46 ± 0.03 |
> |ABCDEFG (SPN)| 0.20 ± 0.02 |0.17 ± 0.02 | 0.28 ± 0.04 |0.55 ± 0.05 |
> |ABCDEFG INTV.| 0.36 ± 0.01 |0.38 ± 0.01 | 0.46 ± 0.10 |0.85 ± 0.02 |
> |ABCDEFG (SPN) INTV.| 0.35 ± 0.01 |0.35 ± 0.02 | 0.51 ± 0.01 | 0.84 ± 0.01|
>
> SHD
> | METHOD | HARD INTVN | SOFT INTVN |SPN HARD | SPN SOFT |
> | :----- | :-----: | :-----: |:-----: | :-----: |
> |RANDOM| 545 ± 46 | 603 ± 31 | 3214 ± 61 | 3233 ± 98 |
> |ABCDEFG| 857 ± 112 | 1121 ± 261| 3067 ± 56 | 2632 ± 217 |
> |ABCDEFG (SPN)| 1076 ± 326 | 1399 ± 342 | 3132 ± 148 |2307 ± 239 |
> |ABCDEFG INTV.| 1659 ± 240 | 1426 ± 346 | 2584 ± 440 |1021 ± 56 |
> |ABCDEFG (SPN) INTV.| 1761 ± 204 | 1516 ± 280 | 2438 ± 187 | 1071 ± 71|

---

> > ### Author Response · Authors · 2025-11-28
> >
> > # Random baseline for F1 and SHD (continued)
> > Table 4 (comparisons with Bayesian methods)
> >
> > F1 score
> > | METHOD | LINEAR| LINEAR SPN | NONLINEAR | NONLINEAR SPN |
> > | :----- | :-----: | :-----: |:-----: | :-----: |
> > |RANDOM| 0.12 ± 0.05 |0.13 ± 0.07 | 0.12 ± 0.04 |0.14 ± 0.03 |
> > |BACADI| 0.18 ± 0.02 |0.22 ± 0.03 | 0.16 ± 0.03 |0.20 ± 0.03 |
> > |DECI| 0.09 ± 0.02 |0.11 ± 0.01 | 0.08 ± 0.01 |0.08 ± 0.02 |
> > |VI-DP-DAG| 0.20 ± 0.04 |0.20 ± 0.03 | 0.13 ± 0.00 |0.21 ± 0.06 |
> > |PRODAG| 0.17 ± 0.01 |0.20 ± 0.02 | 0.16 ± 0.03 |0.23 ± 0.05 |
> > |ABCDEFG| 0.74 ± 0.13 |0.49 ± 0.13 | 0.23 ± 0.31 |0.35 ± 0.24 |
> > |ABCDEFG (SPN)| 0.40 ± 0.03 |0.24 ± 0.06 | 0.13 ± 0.13 |0.30 ± 0.24 |
> >
> > SHD
> > | METHOD | LINEAR| LINEAR SPN | NONLINEAR | NONLINEAR SPN |
> > | :----- | :-----: | :-----: |:-----: | :-----: |
> > |RANDOM| 44.05 ± 2.52  | 51.48 ± 5.20 | 41.36 ± 7.23 |51.33 ± 10.11 |
> > |BACADI| 108.28 ± 0.95 |160.50 ± 1.49 | 109.34 ± 0.82 |178.78 ± 0.85 |
> > |DECI| 37.27 ± 0.76 | 41.89 ± 5.36 | 36.75 ± 4.17 |41.88 ± 4.35 |
> > |VI-DP-DAG| 83.88 ± 4.32 |79.48 ± 1.97 | 86.51 ± 5.35 |79.78 ± 4.28 |
> > |PRODAG| 98.24 ± 1.56 |94.79 ± 1.56 | 81.16 ± 0.60 |88.00 ± 2.52 |
> > |ABCDEFG| 12.74 ± 5.02 |29.25 ± 3.57 | 22.14 ± 5.44 |27.68 ± 0.88 |
> > |ABCDEFG (SPN)| 34.40 ± 1.80 |43.11 ± 2.86 | 30.38 ± 6.78 |34.31 ± 3.87 |
> >
> > # Comparison against Bayesian methods based on MCMC
> > The reviewer is correct that all of the state-of-the-art methods we compared against are based on variational Bayes rather than MCMC. Variational methods are generally much more scalable in high-dimensional and large n settings, which may be why none of these methods used MCMC. Following the reviewer’s suggestion, we compared our model with the MCMC-based method you mentioned (BiDAG) on the 16 node dataset which we listed in Table 4. Our model also outperformed BiDAG.
> >
> > F1 score (MAP)
> > | METHOD | LINEAR | LINEAR SPN |NON LINEAR | NON LINEAR SPN |
> > | :----- | :-----: | :-----: |:-----: | :-----: |
> > |BIDAG| 0.86 ± 0.17 |0.78 ± 0.16 | 0.38 ± 0.30 |0.45 ± 0.20 |
> > |ABCDEFG| 0.90 ± 0.16 |0.71 ± 0.20 | 0.79 ± 0.05 |0.52 ± 0.29 |
> > |ABCDEFG SPN| 0.67 ± 0.02 |0.61 ± 0.16 | 0.70 ± 0.00 |0.49 ± 0.37 |
> >
> > SHD (MAP)
> > | METHOD | LINEAR | LINEAR SPN |NON LINEAR | NON LINEAR SPN |
> > | :----- | :-----: | :-----: |:-----: | :-----: |
> > |BIDAG| 6.67 ± 8.08 |17.00 ± 15.39 | 18.00 ± 6.56 |23.00 ± 3.61 |
> > |ABCDEFG| 4.00 ± 6.93 |18.67 ± 11.93 | 14.00 ± 4.00 |19.67 ± 4.62 |
> > |ABCDEFG SPN| 20.67 ± 4.73 |23.00 ± 9.17 | 23.33 ± 5.51 |22.00 ± 7.00 |
> >
> > # Definition on line 147
> > The reviewer is correct that the joint distribution on L147 should match L137. We intended to use the observational distribution that is implicitly included in the intervention set ($I_1$) in the definition but forgot to complete this part of the definition.
> >
> > # Markov equivalence class
> > Yes, “equivalence class” refers to the interventional Markov equivalence class defined later. We thought the term interventional Markov equivalence class would be too confusing without the definition but we see the reviewer’s point and can change this.
> >
> > #Meaning of m in table 1
> > Yes, m is the number of factors.

---

### Official Review · Reviewer_Vbzh · 2025-11-01

**Soundness:** 3
**Presentation:** 2
**Contribution:** 2
**Rating:** 4
**Confidence:** 3

**Summary:**

This work introduces a scalable causal discovery method that guarantees acyclicity, supports graphs with thousands of nodes, and incorporates interventions even when their targets are unknown. The proposed approach, ABCDEFG, estimates a full posterior distribution over causal structures, whose mode is provably identifiable up to the relevant equivalence class, allowing uncertainty quantification and principled recoverability guarantees. The key innovation lies in representing causal relationships using extended factor graphs, where feature nodes and intervention nodes are connected through auxiliary factor nodes. Building upon the f-DAG framework introduced by Lopez et al. [17], proposed method extends factor graph modeling to handle interventions with unknown targets by introducing factors that capture their influence on affected nodes. Through this extended formulation, the method achieves accurate and scalable distributional estimation of causal DAGs.

**Strengths:**

- The paper establishes clear identifiability guarantees by proving that the mode of the estimated posterior recovers the true causal graph up to the appropriate equivalence class.
- The amortized Bayesian framework enables efficient inference, allowing the method to scale to large graphs and large-scale datasets while maintaining exact acyclicity.

**Weaknesses:**

- The methodological framework is not novel that largely based on prior work, particularly Lopez et al. "Large-Scale Differentiable Causal Discovery of Factor Graphs" (NeurIPS 2022).
- The experimental section would benefit from clearer characterization and comparative analysis of the datasets used. The current presentation lacks a clear description of dataset properties (e.g., underlying causal structure, mechanisms, and complexity), making it difficult to assess how these factors influence performance. Comparative analysis over these factors would help clarify the strengths and potential limitations of the proposed approach in different settings.
- Despite emphasizing scalability as a major advantage, experimental validation in large-scale scenarios is limited. The large-graph evaluation in Section 3.2 is difficult to interpret since results are primarily summarized in text (L429–473). Additionally, the visualization in Figure 2b lacks sufficient explanation, making its practical relevance unclear. It would be valuable to include a more systematic evaluation on real large-scale datasets, such as those used in “Large-Scale Targeted Cause Discovery via Learning from Simulated Data” (TMLR 2025), to more convincingly demonstrate scalability and practical impact.

**Questions:**

- In Appendix Table 14, what is the unit used in the time analysis?
- How does the performance change on simulated data when varying causal structure, mechanisms, and graph complexity?
- On which types of data does the proposed method encounter empirical limitations?
- How does the method compare to other applicable approaches on large-scale single-cell data, and to what extent are the reported performance metrics meaningful from the perspective of biological practitioners?
- What is the practical significance and interpretation of the visualization shown in Figure 2b?

---

> ### Author Response · Authors · 2025-11-28
>
> We thank the reviewer for the thoughtful comments and would like to clarify several points in response.
>
> # Differences from Lopez et al.
> Although our work is based on the factor graph concept from the DCDFG paper, we have made four significant contributions that we think merit publication.
>
> (1)  We learn causal graphs that are acyclic by construction, avoiding the need for expensive approximate acyclicity constraints. The original DCDFG often produces graphs that contain cycles, violating the assumptions of their theoretical framework.
>
> (2)  We model interventions with unknown targets. The original DCDFG cannot do this. This is important for biology applications because interventions like drug and growth factor treatments do not have known targets.
>
> (3) We develop a variational Bayesian inference framework that learns distributional estimates and achieves much better performance compared to the DCDFG model.
>
> (4) We prove that our approach recovers the true causal graph up to an interventional Markov equivalence class, whereas DCDFG has no theoretical guarantees.
>
> # Description of dataset properties and evaluation of how causal structure, mechanisms, and complexity affect performance
> Thank you for this suggestion. These are admittedly complex datasets, but we did perform comparative analyses over key factors influencing performance of causal inference.
> We evaluated linear vs non-linear causal effects: whether the child node is sampled using a linear or non-linear function from the parent nodes.
> We sampled causal edges with independent vs. jointly distributed probabilities. This refers to the process of modelling edge probability. Independent means we use the erdos-renyi graph and sampled the edge probability independently. We also modeled the edge probability jointly through a sum-product network (SPN), hence the edge probability is not independent here, and this type of SPN dataset was not used by previous work.
> We evaluated hard vs. soft interventions: we tested whether to completely remove the parent dependence (hard) or retain some parent dependence while modifying the distribution in the SEM structure (soft).
> We showed part of the comparisons from different types of datasets in the main text. For example, in Table 2 we compared score-based methods on non-linear targeted datasets in factor graph (Independent edge probability) or SPN factor graph (jointly distributed edge probabilities) with hard and soft interventions. And Table 3 is the performance on non-linear untargeted datasets. In Table 4, we compared the performance with Bayesian methods on hard targeted interventions on factor graph and SPN factor graph. We put other results in the appendix, where we also simulated the number of nodes (from 100 to 500) and also the sparsity of the graphs. All of these results suggest our model outperformed other methods in most dataset settings, especially for SPN factor, where most methods do not perform well.
>
> # Experimental validation in large-scale scenarios is limited
> We ran our approach on a large-scale real dataset and evaluated the recall of known causal edges and ability to predict gene expression. We think this highlights a key advantage of our approach, because none of the other methods we compared against in the simulated data experiments were able to perform this task. These results are summarized in L429-473. Is the reviewer’s concern that the dataset was not large enough? Our approach certainly scales to much larger datasets; we have additional results on real-world datasets with > 300,000 cells.
>
> # Systematic evaluation on real large-scale datasets as in TMLR 2025 paper
> Following the reviewer’s suggestion, we followed the evaluation in the reference TMLR paper by using the string database to show how many correlations between genes are identified by different models. Here, since the graphs output from different models are directed, we compare two situations: (1) when a gene functions as upstream gene, how many genes from the string database have correlation with the target gene are overlapped from the graph output. (2) when a gene functions as a downstream gene, how many regulator genes from the string database have correlation with the target gene are overlapped from the graph output. We compared SDCD, DCDFG and our models for this evaluation, and overall, our model output a higher number of genes which have correlation support from the string database.
> | METHOD |upstream | downstream |
> | :----- | :-----: | :-----: |
> |DCDFG| 16.76 ± 32.66 | 15.34 ± 18.44 |
> |SDCD| 0.22 ± 1.49 | 4.56 ± 6.00 |
> |ABCDEFG| 24.58 ± 33.69 | 24.91 ± 28.37 |
> |ABCDEFG (SPN)| 23.01 ± 28.97 | 27.32 ± 30.97 |

---

> > ### Author Response · Authors · 2025-11-28
> >
> > # Appendix Table 14 units
> > The unit here is seconds.
> >
> > # On which types of data does the proposed method encounter empirical limitations?
> > Based on our experiments, our model has slightly lower performance compared with the SDCD on factor graphs as shown in Table 2,4, 6 and 7
> >
> > # Comparison to other applicable approaches on large-scale single-cell data
> > This is a key benefit of our approach. There are no other approaches that can perform this type of analysis on large-scale single-cell datasets with interventions lacking known targets. DCDI can in theory work in this setting, but it is too slow to run on large-scale real datasets such as the one we analyzed here. To perform some level of comparison with previous methods, we had to treat the data as observational data, and compare the performance in different aspects like data reconstruction. In lines 427-474 of our paper, we described the identification of gene ontology related genes from our model (intervention target identification). We also outperform a baseline. This intervention target identification task is biologically meaningful, as it quantifies the intervention targets through how many genes are identified by our model for a specific biological process (go term). To compare with other models on this real dataset, we compared the data reconstruction quality with DCDFG and SDCD models. These comparisons are biologically meaningful from the perspective of practitioners.
> >
> > # Significance and interpretation of Fig. 2b
> > Thanks for giving us a chance to clarify. Here is a more clear description of Figure 2b:
> > The pentagons represent perturbations (intervention with unknown targets), and the circles represent causal graph factors. The following types of directed edges are learned from our model: intervention to latent factor, gene to latent factor, and latent factor to gene. From this panel, we want to show the following points:
> >
> > (1) Our model learns perturbation-specific responses through the representation by factors. For example, our model learned the edges from bmp4 and bmp7 perturbations towards factor 2, and through our gene ontology analysis, factor 2 is enriched in terms like response to BMP, which suggests this factor correctly identifies genes in the BMP signaling pathway.
> >
> > (2) Our model learns the regulation relationships between upstream and downstream genes. For example, we colored the text of some genes that have edges from genes to factors in blue in this panel, and we can see most of the genes are transcription factors. This makes sense biologically, because transcription factors are the most common type of upstream regulators in gene regulatory networks. Similarly, we colored the text of genes which have edges from factors to genes in orange. Many of these genes coding proteins like structural proteins and enzymes which usually serve as the terminal effectors of a biological process.
> > 	We also included this figure simply because we think that it helps to communicate the type of causal graph (extended factor graph) learned by our approach and its interpretation in terms of biological mechanisms.

---

### Meta-Review · Area_Chair_a4Ky · 2026-01-08

**Summary:**

- **Summary of the paper**: This paper proposes a scalable causal discovery method, Amortized Bayesian Causal Discovery of Extended Factor Graphs (ABCDEFG), that can scale graphs with thousands of nodes and incorporate intervention with unknown targets. The core idea of ABCDEFG is to represent causal relationships using extended factor graphs. The proposed method demonstrates its effectiveness on a large-scale dataset, recovering both known and unknown interactions.
- **Suggested decision for this paper:** Given the reviews and rebuttals, the AC determines this paper as a borderline paper. Even if the reviewers had participated in the discussion, it is judged that the motivation would still have been difficult to accept or understand, and unfortunately, this paper is given a reject decision.

**Reviewer Concerns:**

- Short summary of the reviewers' concerns:
	- `Reviewer Vbzh`: (score: 4) Lack of methodological novelty largely based on previous work, f-DAG (Lopez et al, NeurIPS 2022). Unclear presentation of experiments. Lack of evaluation on a real large-scale dataset. Unclear writing.
	- `Reviewer 97Bh`: (score: 4) Practicalness. Unclear presentation of experiments. Unclear writing.
	- `Reviewer e9Uu`: (score: 2) Unclear writing, unclear contribution compared to f-DAG.
	- `Reviewer 7yqf`: (score: 6) Unclear motivation for extending factor graphs
- Overall summary: the reviewers raised concerns of lack of novelty (`Reviewer Vbzh`), unclear motivation, especially beyond f-DAG (`Reviewer e9Uu`, `Reviewer 7yqf`), unclear writing and presentation of experiments (`Reviewer Vbzh`, `Reviewer 97Bh`, `Reviewer e9Uu`, `Reviewer 7yqf`)

**Reviewer Scores:**

Given the reviews, rebuttal, and discussion, the AC predicts the revised score if the authors and reviewers had been able to participate in the discussion fully.
- `Reviewer Vbzh`: (score: 4) -> (score: 4)
- `Reviewer 97Bh`: (score: 4) -> (score: 4)
- `Reviewer e9Uu`: (score: 2) -> (score: 4)
- `Reviewer 7yqf`: (score: 6) -> (score: 6)

---

### Decision · Program_Chairs · 2026-01-26

Reject